# FLAM: Frame-Wise Language-Audio Modeling

**Yusong Wu** [1 2]   **Christos Tsirigotis** [2]   **Ke Chen** [1]   **Cheng-Zhi Anna Huang** [3]   **Aaron Courville** [2 4]   **Oriol Nieto** [1]
**Prem Seetharaman** [1]   **Justin Salamon** [1]

## Abstract

Recent multi-modal audio-language models (ALMs) excel at text-audio retrieval but struggle with frame-wise audio understanding. Prior works use temporal-aware labels or unsupervised training to improve frame-wise capabilities, but they still lack fine-grained labeling capability to pinpoint when an event occurs. While traditional sound event detection models can precisely localize events, they are limited to pre-defined categories, making them ineffective for real-world scenarios with out-of-distribution events. In this work, we introduce FLAM, an open-vocabulary contrastive audio-language model capable of localizing specific sound events. FLAM employs a memory-efficient and calibrated frame-wise objective with logit adjustment to address spurious correlations, such as event dependencies and label imbalances during training. To enable frame-wise supervision, we leverage a large-scale dataset with diverse audio events, LLM-generated captions and simulation. Experimental results and case studies demonstrate that FLAM significantly improves the open-vocabulary localization capability while maintaining strong performance in global retrieval and downstream tasks.

## 1. Introduction

Multi-modal contrastive models, such as Vision–Language Models (VLMs) (Radford et al., 2021; Zhai et al., 2023) and Audio–Language Models (ALMs) (Elizalde et al., 2023; Wu* et al., 2023), effectively learn open-vocabulary representations that enable strong retrieval, understanding (Liu et al., 2024), and text-conditioned generation (Ramesh et al.,

2022; Liu et al., 2023a; Evans et al., 2024). In the audio domain, ALMs like CLAP (Elizalde et al., 2023; Wu* et al., 2023) learn instance-level alignments between audio and text. However, these global embeddings cannot precisely localize the temporal boundaries of specific acoustic events.

Enabling local alignment between audio frames and text would significantly benefit applications like audio content search and event detection, allowing users to pinpoint exactly when a described sound occurs. However, unlike the image domain—where large datasets (Schuhmann et al., 2022) and granular segmentation labels (Kirillov et al., 2023) are more readily available—audio frame-level annotations paired with text are exceedingly scarce. Existing Sound Event Detection (SED) datasets (Serizel et al., 2020) often have a limited, fixed vocabulary and remain relatively small in size due to the significant human effort required for annotation. Although self-supervised approaches (Xu et al., 2021; 2024; Li et al., 2024) attempt to learn local alignment from audio-text pairs, the overall data volume remains modest relative to other domains, limiting their scalability.

In this paper, we present **FLAM** (**F**rame-Wise **L**anguage–**A**udio **M**odeling), an ALM model designed for frame-level open-vocabulary SED. Unlike standard ALMs, FLAM's audio encoder provides both a global sample-level embedding and a sequence of frame-level embeddings. By matching each frame embedding to text embeddings, FLAM can detect not only *whether* a sound event is present but also *when* it occurs within the clip.

We train FLAM with a frame-level contrastive objective and incorporate logit adjustment techniques (Menon et al., 2021; Tsirigotis et al., 2023) to deal with the spurious correlations of frame labels during training. Then, we propose a memory-efficient training strategy, which handles large batches of frame-wise data without compromising computational feasibility. To overcome the lack of frame-level audio-text annotations, we build a large-scale data augmentation pipeline that synthesizes 10-second audio mixtures from text-labeled acoustic events. This approach automatically re-labels event boundaries, creating a one-million-sample dataset of diverse, open-vocabulary SED examples.

---

[1]Adobe Research [2]Mila - Quebec AI Institute, Université de Montréal [3]Massachusetts Institute of Technology [4]Canada CIFAR AI Chair. Correspondence to: Yusong Wu <wu.yusong@mila.quebec>, Justin Salamon <salamon@adobe.com>.

*Proceedings of the $42^{nd}$ International Conference on Machine Learning*, Vancouver, Canada. PMLR 267, 2025. Copyright 2025 by the author(s).

Our key contributions are as follows:

- **Frame-Level Open-Vocabulary SED:** We introduce FLAM, extending ALMs to produce both sample-level and frame-level representations for open-vocabulary event detection.

- **De-biased Frame-level Contrastive Learning:** We develop a frame-level contrastive objective that includes a bias correction term and an unbiased event classifier, effectively handling label imbalance in SED training (Figure 1).

- **Scalable Data Augmentation:** We propose a pipeline that synthesizes audio mixtures with precise event boundaries from text-labeled corpora, yielding a *large-scale* (1M samples) open-vocabulary SED dataset.

- **State-of-the-Art Performance:** FLAM outperforms prior self-supervised approaches on both traditional closed-set and open-set SED (Fig. 2, Table 1), while preserving the strong retrieval (Table 2) and zero-shot classification (Table 3).[1]

## 2. Preliminaries

As a multi-modal representation learning paradigm, contrastive learning methods introduce instance-level classification tasks using paired observations found in batches at each training step. Specifically in CLIP (Radford et al., 2021), which is also adopted by ALMs such as CLAP (Elizalde et al., 2023; Wu* et al., 2023), a contrastive learning task is formed so that individual samples from one modality can be classified against alternative samples in a batch using their associated observations from the other modality, and vice versa. In more detail, for audio-language contrastive learning, suppose that we sample a batch $\mathcal{B} = \{(X_i, Y_i)\}_{i=1}^{B}$ from the training dataset, which contains audio samples, $X_i$, paired with text descriptions, $Y_i$. An audio encoder $f^a$ maps an audio sample $x$ to a $d$-dimensional embedding, which is afterwards $L_2$-normalized to produce $\mathbf{e}^a \in \mathbb{R}^d$. Likewise, a text encoder $f^t$ maps text descriptions $y$ to embeddings which after $L_2$-normalization we call $\mathbf{e}^t \in \mathbb{R}^d$. The encoders are based on feature extractors, $E^a$ and $E^t$ respectively, followed by Multi-Layer Perceptrons (MLP). In brief, we get instance-level embeddings for an audio sample $x$ and a text description $y$ by

$$\mathbf{e}^a(x) = \frac{f_a(x)}{\|f_a(x)\|_2}, \quad \mathbf{e}^t(y) = \frac{f_t(y)}{\|f_t(y)\|_2},$$

and we type $\mathbf{e}_i^a = \mathbf{e}^a(X_i)$ to mean the embedding of an audio $X_i$ in a batch, and similarly $\mathbf{e}_i^t = \mathbf{e}^t(Y_i)$ the embedding of a text description $Y_i$.

---

[1]Detailed results with real-world sound event detection examples are shown at: https://flam-model.github.io/

Given these instance-level embeddings, CLIP models classify pairs of observations in a batch $\mathcal{B}$ by minimizing an InfoNCE (van den Oord et al., 2018) objective, $\mathcal{L}_{\text{CLIP}} =$

$$-\frac{1}{2B} \sum_{i=1}^{B} \left( \log \frac{e^{\alpha \mathbf{e}_i^a \cdot \mathbf{e}_i^t}}{\sum_{j=1}^{B} e^{\alpha \mathbf{e}_i^a \cdot \mathbf{e}_j^t}} + \log \frac{e^{\alpha \mathbf{e}_i^a \cdot \mathbf{e}_i^t}}{\sum_{j=1}^{B} e^{\alpha \mathbf{e}_j^a \cdot \mathbf{e}_i^t}} \right), \quad (1)$$

where $\alpha$ is a logit scale, $\alpha = e^{\alpha'}$, controlled by a trainable parameter $\alpha'$. We would like the dot product between embeddings, $\mathbf{e}_i^a \cdot \mathbf{e}_j^t$, to be high for positive pairs, where $i = j$, and low for negative pairs, for which $i \neq j$.

Another form of instance-level contrastive representation learning method is explored by Zhai et al. (2023, SigLIP). In their work, pairs of observations are directly classified as positive or negative. Instead of an InfoNCE loss, Zhai et al. (2023) minimize a binary cross-entropy loss $\mathcal{L}_{\text{SigLIP}} =$

$$-\frac{1}{B} \sum_{i=1}^{B} \sum_{j=1}^{B} \log \sigma \left( z_{i,j} \left( \alpha \mathbf{e}_i^a \cdot \mathbf{e}_j^t + \beta \right) \right), \quad (2)$$

where $z_{i,j} = 1$ if $i = j$ and $z_{i,j} = -1$ if not, and $\sigma$ is the logistic function. As before $\alpha > 0$ is a trainable logit scale, and $\beta$ is a trainable logit bias. Proper initialization for the logit bias is crucial for effective model training, mitigating the effects of $1 : B - 1$ label imbalance between positive and negative pairs. If an appropriate logit bias did not exist, a model could trivially predict negative values for $\mathbf{e}_i^a \cdot \mathbf{e}_j^t$ for all $i, j \in [B]^2$, thus impeding learning in large batch sizes. In contrast, offsetting logits with an appropriate bias, which captures the training artifact of the marginal statistic on $z$ labels, enables learning high $\mathbf{e}_i^a \cdot \mathbf{e}_j^t$ values when $z_{i,j} = 1$ and low values when $z_{i,j} = -1$. We include details in §C.1.

## 3. Methodology

Open-vocabulary SED aims to locate text descriptions of acoustic events in an audio signal. Figure 1 provides a high-level comparison between traditional contrastive ALMs and our proposed FLAM approach for frame-level event detection. In this work, we train our model directly on a labeled open-vocabulary SED dataset. At each training step of our method, we sample a batch from the dataset $\mathcal{B}_{\text{SED}} = \{(X_i, \{(Y_{i,k}, Z_{i,k}^{loc})\}_{k=1}^{K_i})\}_{i=1}^{B}$. The batch contains audio clip samples $X_i$, along with a *variable number* $K_i$ of positive text descriptions $Y_{i,k}$ that correspond to the acoustic events present in each audio clip. Each text description is accompanied by frame-wise labels $Z_{i,k}^{loc} \in \{-1, 1\}^L$ indicating *when* in the audio each event occurs. When $Z_{i,k,l}^{loc} = 1$, it means that the acoustic event described by $Y_{i,k}$ is present in $X_{i,l}$, frame $l \in [L]$ of audio clip $X_i$. Conversely, $Z_{i,k,l}^{loc} = -1$ indicates that the acoustic event is not audible at frame $X_{i,l}$. We construct SED data by synthesizing audio mixtures from a larger audio-text corpus, and we describe the data augmentation process in §4.

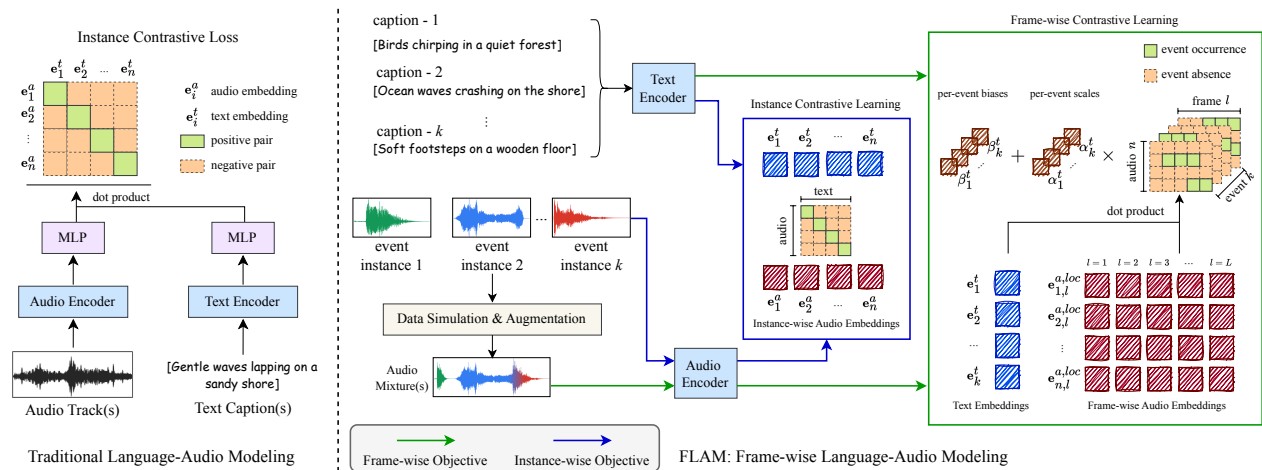

*Figure 1.* Comparison of traditional audio-language contrastive modeling (**left**) with the proposed Frame-wise Language-Audio Modeling (FLAM, **right**). Traditional models generate global embeddings for audio-text pairs and optimize an instance-wise contrastive objective. In contrast, FLAM integrates both instance-wise and fine-grained frame-wise contrastive objectives. Audio mixtures are synthesized from event instances, paired with text descriptions, and processed to produce frame-wise embeddings. Frame-wise contrastive learning explicitly aligns individual audio frames with textual event descriptions using event-specific scales and biases, enabling precise temporal localization of sound events.

Traditional SED targets a fixed set of classes, predicting a binary label per time frame. In contrast, we propose open-vocabulary SED in this work as detecting the temporal boundary of *any* event described by some text $y$. To this end, we propose that, given an audio sample x, the audio encoder $f^{a,loc}$ outputs frame-level representations in $\mathbb{R}^{L \times d}$, which we consequently $L_2$-normalize (Figure 1) along the embedding dimension,

$$\mathbf{e}^{a,loc}(x)_l = \frac{f^{a,loc}(x)_l}{\|f^{a,loc}(x)_l\|_2} \in \mathbb{R}^d, \quad l \in [L] \quad (3)$$

consisting of $d$-dimensional embeddings for each frame out of total $L$ frames. Unchanged from the instance-level setting, the text encoder $f_t$ yields a single embedding $\mathbf{e}^t(y) = \frac{f^t(y)}{\|f^t(y)\|_2} \in \mathbb{R}^d$ for any text query $y$. For notational convenience in the batch setting, we write $\mathbf{e}^{a,loc}_{i,l} = \mathbf{e}^{a,loc}(X_i)_l$ to represent the embedding of frame $l$ of an audio sample $X_i$.

Unlike contrastive ALMs which are usually concerned only with instance-level *rankings* via a comparison of dot products of text query and audio clip embeddings, open-vocabulary SED requires 1) detection with unlimited open-set language prompts, and 2) calibrated *probabilities* for each frame and event. This is because each frame can contain a variable number of active events, including none. To detect temporal occurrence of acoustic events, we propose to construct a classifier which takes a temporal audio embedding and a text embedding, and detects whether an event $y$ occurs in frame $l \in [L]$ of audio $x$.

The reason for proposing this formulation is twofold. First, it can efficiently leverage current contrastive ALMs. Con-

trastive ALMs often produce temporal representations, $\mathbf{e}^{a,loc}(\cdot)$, in the second-to-the last layer, which we can use to obtain a global representation, $\mathbf{e}^a(\cdot) = \frac{1}{L} \sum_{l \in [L]} \mathbf{e}^{a,loc}(\cdot)_l$, by averaging across the frame dimension (Chen et al., 2022). Thus, our formulation can be built upon current ALMs with minimal computation overhead on model inference. Second, our formulation is computationally efficient at inference time. Compared to an alternative formulation which directly outputs events matching given both audio and event text query, our formulation allows us to precompute temporal audio representations $\mathbf{e}^{a,loc}(\cdot)$ and only compute text embeddings when a new prompt is given.

Figure 1 illustrates the key differences between traditional audio-language modeling and our proposed FLAM framework. Traditional approaches (left) produce global embeddings and optimize instance-level alignment between audio and text pairs. In contrast, FLAM (right) leverages both global and frame-level embeddings to enable precise temporal localization. Specifically, FLAM synthesizes audio mixtures from diverse events, creating temporally aligned frame-level embeddings. These embeddings are explicitly aligned with textual event descriptions through frame-wise contrastive learning, augmented by event-dependent scaling and biases. This dual objective structure allows FLAM to not only detect *whether* events occur but precisely localize *when* they occur in audio clips.

### 3.1. Robust Training and Inference

We propose to formulate the open-vocabulary SED as a **contrastive objective** based on **binary classification**. Given

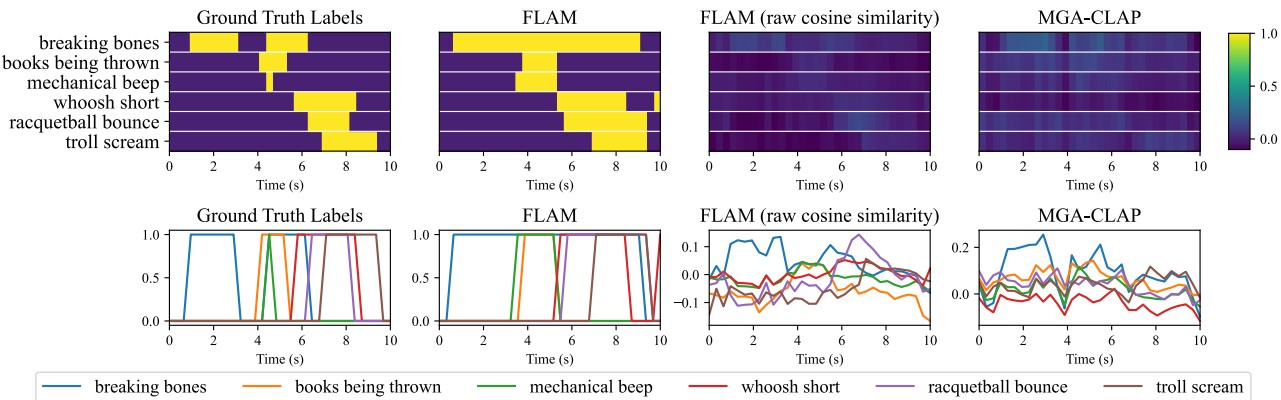

*Figure 2.* Example outputs from FLAM and MGA-CLAP on open-vocabulary sound-event-detection example (top row) with detailed value at each frame (bottom row). **FLAM**, trained with event-dependent logit adjustment, converts **raw cosine similarity** into calibrated predictions. By contrast, the unsupervised MGA-CLAP model produces less accurate results and cannot be calibrated from its output.

a batch $\mathcal{B}_{\text{SED}}$ randomly sampled from the training set, we collect all the event descriptions found in the batch $\mathcal{Y} = \bigcup_{i \in [B]} \{Y^{(i,k)}\}_{k=1}^{K_i}$ with $|\mathcal{Y}| = K$ elements, and re-define frame-wise labels on the union of description for all $i \in [B], k' \in [K], l \in [L]$, as $z_{i,k',l} = 1$ if $\mathcal{Y}_{k'} = Y^{(i,k)}$ and $Z_{i,k,l}^{loc} = 1$ for some $k \in [K_i]$, otherwise $z_{i,k',l} = -1$. In essence by computing the union of event descriptions, we deduplicate potential overlapping acoustic events present in different audio samples in the batch while maintaining consistency with the dataset about positive audio frames with respect to acoustic event descriptions. In addition, we assume that all frames of an audio sample $X_i$ are negative with respect to an acoustic event description $y$ which does not appear in the dataset for that audio sample, $y \notin \{Y_{i,k}\}_{k=1}^{K_i}$. We propose to minimize the open-vocabulary SED objective $\mathcal{L}_{\text{SED}} =$

$$-\frac{1}{BKL} \sum_{i=1}^{B} \sum_{k=1}^{K} \sum_{l=1}^{L} \log \sigma \left( z_{i,k,l} \, h(X_i, l, \mathcal{Y}_k) \right), \quad (4)$$

where $h(x, l, y)$ is a logit function we model as

$$h(x, l, y) = \alpha^t(y) \, \mathbf{e}^{a,loc}(x)_l \cdot \mathbf{e}^t(y) + \beta^t(y) \quad (5)$$

with $\alpha^t(\cdot) > 0$ being a text-dependent logit scale and $\beta^t(\cdot)$ a text-dependent logit bias. Note that we slightly abuse notation so that $\mathbf{e}_k^t = \frac{f^t(\mathcal{Y}_k)}{\|f^t(\mathcal{Y}_k)\|_2}$ in Figure 1.

Compared to $\mathcal{L}_{\text{SigLIP}}$ in Eq. 2, our loss in Eqs. 4-5 is still a binary cross-entropy classification objective, which acts at the frame-level of an audio instead of the instance-level, and aims to learn $p_{\text{data}}(z = 1 \mid x, l, y)$ via $\sigma(h(x, l, y))$. Moreover, in contrast to SigLIP, our logit scale and bias are conditioned on event descriptions. As we show in the following sections, this allows us to model label-imbalance structures on a finer level which we argue it is important for open-vocabulary SED.

### 3.2. Logit Adjustment for Event-dependent Imbalances

From the binary classification perspective of our objective $\mathcal{L}_{\text{SED}}$ (§3.1), labels $z_{i,k,l}$ are highly imbalanced, as the large majority of frame–text pairs are negative. Beyond the dependence on the batch size, which increases the number of negative frame-text pairs as we have discussed in §2 about SigLIP, we need to consider dependencies of positive frame-text pairs on the event descriptions present in our training dataset. For example, "thunder" may occur infrequently and have a short duration in our dataset, whereas "rain falls" may appear more frequently and persist for a longer time. It is important to counteract such idiosyncrasies of our training dataset, if we would like our final model to perform robustly across event descriptions in the open-vocabulary case.

To avoid overfitting to dataset-specific priors and the particular choice of batch size, we introduce a text-dependent logit bias term $\beta^t(\cdot)$ in the logit function of Eq. 5. Prior work on long-tailed multi-class classification (Menon et al., 2021), and more recently on training classifiers under spurious correlations (Liu et al., 2023b; Tsirigotis et al., 2023), has developed logit adjustment techniques to handle distribution shifts due to training data biases. Here, we adapt logit adjustment to binary classification in order to deal with the aforementioned label imbalance challenges.

**Inference** Our goal is to robustly determine whether each frame $(x, l)$ contains a given event $y$. We seek a classifier that treats positive and negative predictions equally regardless of $y$. As we show in §C.3, the Bayes-optimal classifier, under our working hypotheses, is given by

$$z^*(x, l, y) = \arg \max_{z \in \{-1,1\}} \frac{p_{\text{data}}(z \mid x, l, y)}{p_{\text{data}}(z \mid y)}. \quad (6)$$

Since $z \in \{-1, 1\}$, taking the $\arg\max$ corresponds to checking whether $p_{\text{data}}(z = 1 \mid x, l, y) > p_{\text{data}}(z =$

*Table 1.* Sound event detection performance on synthetic open-vocabulary SED (Held-out, ASFX-SED) and traditional closed-set SED dataset (DESED, MAESTRO, Audioset-strong, UrbanSED). FLAM produces more accurate sound event detection compared with existing models on both open-vocabulary and closed-set SED datasets. Bold numbers shows the best performance across all models. MGA-CLAP* is MGA-CLAP trained on our data.

| Model | Held-out | ASFX-SED | DESED | | MAESTRO | Audioset-S | | UrbanSED | |
|---|---|---|---|---|---|---|---|---|---|
| | AUROC | AUROC | PSDS | AUROC | MPAUC | PSDS | AUROC | PSDS | AUROC |
| FLAM-Global | 67.76 | 65.14 | 7.09 | 85.52 | 51.13 | 1.11 | 82.54 | 0.82 | 67.39 |
| FLAM (proposed) | **91.0** | **81.23** | 9.37 | **91.66** | **56.97** | **11.16** | **94.76** | **29.52** | **93.62** |
| MGA-CLAP* | 74.17 | 69.56 | **14.72** | 89.28 | 52.50 | 1.24 | 79.12 | 6.42 | 78.22 |
| MGA-CLAP (reported) | - | - | **26.4** | - | - | 10.1 | - | 8.7 | - |

$1 \mid y$). In other words, $p(z = 1 \mid y)$ acts as the classification threshold for the posterior $p(z = 1 \mid x, l, y)$. To achieve a fixed boundary at 0.5, independent of the event description $y$, we define

$$s(x, l, y) = \frac{p_{\text{data}}(z = 1 \mid x, l, y)}{p_{\text{data}}(z = 1 \mid x, l, y) + p_{\text{data}}(z = 1 \mid y)}. \quad (7)$$

A value of $s(x, l, y) > 0.5$ indicates a positive detection. Furthermore, if $\beta^*(y) = \log \frac{p_{\text{data}}(z=1|y)}{p_{\text{data}}(z=-1|y)}$ is sufficiently negative, the ratio in Eq. 7 can be approximated by

$$s(x, l, y) \approx \sigma\left(\log \frac{p_{\text{data}}(y \mid x, l)}{p_{\text{data}}(y)}\right). \quad (8)$$

**Logit Bias Training** Assuming that our training and model hypothesis are able to attain the minimum of our $\mathcal{L}_{\text{SED}}$ loss in Eq. 4, our model learns

$$h^*(x, l, y) = \log \frac{p_{\text{data}}(y \mid x, l)}{p_{\text{data}}(y)} + \beta^*(y), \quad (9)$$

as we show in §C.2. We observe that if we have the logit bias $\beta^t(y)$ in Eq. 5 approximate $\beta^*(y)$, then we effectively absorb the spurious train-time statistical relationships and enable $\alpha^t(y) \, \mathbf{e}^{a,loc}(x)_l \cdot \mathbf{e}^t(y)$ in Eq. 5 to approximate the robust logit $\log \frac{p_{\text{data}}(y|x,l)}{p_{\text{data}}(y)}$ of the classifier in Eq. 8.

However, computing $\beta^*(y)$ for every possible text prompt $y$ is generally intractable. We therefore approximate this quantity via independently training $\beta^t$ as an auxiliary classifier, implemented via a lightweight MLP appended to the text feature extractor, $\beta^t(y) = \text{MLP}^p\big(E^t(y)\big)$. We train the MLP to minimize the loss $\mathcal{L}_p =$

$$-\frac{1}{K} \sum_{k=1}^{K} \Big[ \bar{z}_k \log \sigma\big(\beta^t(\mathcal{Y}_k)\big) + (1 - \bar{z}_k) \log \sigma\big(-\beta^t(\mathcal{Y}_k)\big) \Big], \quad (10)$$

where $\bar{z}_k = \frac{1}{BL} \sum_{i \in [B], l \in [L]} 0.5 \, (z_{i,k,l} + 1) \in [0, 1]$, the average label for prompt $\mathcal{Y}_k$ across all frames $l$ and clips $i$ in the current batch. This approach introduces minimal computational overhead and leverages the language understanding of the text encoder. To prevent SED objective interfere with bias estimation, we stop gradients propagate from the $\mathcal{L}_{\text{SED}}$ in Eq. 4 to the MLP of classifier $\beta^t$. We train the text-dependent logit scale $\alpha^t$ in similar manner where another MLP appended to text feature extractor, giving $\alpha^t(y) = \text{MLP}^\alpha(E^t(y))$. Different in per-text bias, we update $\text{MLP}^\alpha$ via $\mathcal{L}_{\text{SED}}$ in Eq. 4.

Finally, we have experimentally found that adopting a per-text logit scale, which is trained by minimizing $\mathcal{L}_{\text{SED}}$ of Eq. 4, to be additionally beneficial for downstream open-vocabulary SED.

### 3.3. Memory-Efficient Training

Computing the full $\mathcal{L}_{\text{SED}}$ across all $B \times K \times L$ frame-text pairs can be prohibitively memory-intensive. To address this challenge, we adopt a chunked approach inspired by SigLIP (Zhai et al., 2023) that avoids gathering all embeddings on a single GPU. Specifically, each of the $N^{\text{GPU}}$ GPUs processes its local subset of audio frames and text prompts to compute pairwise losses. Since each audio clip may contain a varying number of events, we allocate text slots equal to five times the number of audio clips in a batch, padding any unused audio or text entries with placeholders. Next, we pass the text embeddings (and associated masks) to the next GPU in a ring. After $N^{\text{GPU}} - 1$ transmissions, each GPU has accumulated all cross-device loss terms without centralized data collection. This strategy enables large-batch training while respecting single-GPU memory constraints.

## 4. Dataset and Data Augmentation

FLAM is trained on two data sources: (1) a large-scale audio–text corpus, similar to those used by contrastive ALMs, and (2) an open-vocabulary SED dataset synthesized by inserting one or more sound events (and their captions) into 10-second background clips. Below, we first describe the audio–text corpus (§4.1), then explain the augmentation procedure (§4.2).

*Table 2.* Recall performance of text to audio (T2A) and audio to text (A2T) retrieval. FLAM has comparable Audio-text retrieval performance evaluated on ASFX, Clotho and Audiocaps datasets. Bold numbers shows the best performance across models trained on our dataset, while underlined numbers indicates best performance among all models. MGA-CLAP* is MGA-CLAP trained on our data.

| Model | Dataset | ASFX | | | | Clotho | | | | AudioCaps | | | |
|---|---|---|---|---|---|---|---|---|---|---|---|---|---|
| | | T2A | | A2T | | T2A | | A2T | | T2A | | A2T | |
| | | R@1 | R@5 | R@1 | R@5 | R@1 | R@5 | R@1 | R@5 | R@1 | R@5 | R@1 | R@5 |
| FLAM - Global | | **4.4** | 14.8 | **4.0** | 13.8 | **14.3** | **35.8** | 17.9 | 39.8 | 36.0 | **70.5** | 46.1 | **78.6** |
| FLAM | FLAM-Collection | **4.4** | **15.2** | 3.9 | **13.9** | 13.8 | 33.2 | 16.7 | **42.2** | 32.1 | 64.8 | 43.3 | 75.0 |
| MGA-CLAP* | | 3.9 | 14.8 | 3.9 | 13.8 | 13.4 | 30.3 | **18.7** | 39.1 | **36.7** | 69.9 | **47.2** | 78.3 |
| **Reported Performance from Prior Studies (but trained on different datasets)** | | | | | | | | | | | | | |
| LAION-CLAP | LAION-Audio-630K | 2.0 | 7.6 | 1.6 | 6.0 | 16.1 | 38.3 | 22.7 | 48.5 | 36.1 | 71.8 | 46.8 | 82.9 |
| CompA | CompA-Collection | - | - | - | - | 16.8 | 43.5 | 23.9 | 50.7 | 36.1 | 78.6 | 47.8 | 83.5 |
| MGA-CLAP | WavCaps | 2.3 | 8.3 | 2.0 | 7.4 | 20.4 | 46.0 | 25.3 | 51.2 | 41.8 | 76.1 | 54.4 | 83.6 |

### 4.1. Audio–Text Dataset

We gather a large mix of licensed proprietary sound effect datasets and publicly available CC-licensed general audio datasets, consisting of approximately 1.1M audio samples with corresponding metadata. Sound-effect data typically contain a single clean event accompanied by tags or captions, while general-audio datasets are noisier and may contain multiple events that are not fully described. Because the sound-effect clips are generally clean, they are well-suited for insertion into diverse backgrounds without significant interference. We prompt Mixtral (Jiang et al., 2024) with the file name, tags, and any available textual descriptions to generate captions of lengths in 2-13 words.

### 4.2. SED Data Augmentation

**Event and Background Audio Filtering**   We divide the corpus into two categories: *sound events*, which are shorter than 10 seconds and do not contain the keyword "ambiance", and *background audio*, which lasts at least 10 seconds and includes the keyword "ambiance".

**Event Selection and Placement**   To create a 10 s training mixture, we randomly pick one background clip and sample $N \sim \mathcal{U}(1, 10)$ events. In 80% of cases, events are drawn from sound effect datasets; in 20%, from general audio datasets. Each event is placed at a random start time, with at most three events overlapping.

**Splitting and Repetition**   A naive placement procedure risks biasing the model to single instances of each event whereas real-world scenario often seek to detect events that are repeated or fragmented in time. To reflecting realistic event occurrence, each sound event is split into $\mathcal{U}(2, 3)$ segments (with each segment at least 0.5 s long) in 10% of the cases, and repeated $\mathcal{U}(2, 3)$ times in other 10% of the cases.

**Random Loudness and Mixing**   An offset is sampled from $\mathcal{U}(6, 30)$ dB for each event relative to the background.

We then randomly place each event into random position, allowing a maximum of 3 concurrent sound events. When mixing audio, we apply a 10 ms fade-in/out for each event before mixing to ensure natural onsets and offsets.

**RMS-Based Boundary Correction**   To reduce label noise from silence in events, we compute the A-weighted RMS loudness over each event and classify frames below $-70$ dB as inactive. We also include a smoothing step to avoid rapid label fluctuations (see § C.4).

## 5. Experiments

We pursue four main research questions in this work: (1) How does FLAM perform on both open-vocabulary and closed-set SED tasks? (2) How does FLAM perform on standard audio–text retrieval benchmarks? (3) How does FLAM perform on downstream tasks typically used to evaluate contrastive ALMs? (4) How effective is our proposed design of combining frame-wise and global contrastive learning?

### 5.1. Training Setup

Our model architecture follows the LAION-CLAP framework. The audio encoder $E^a(\cdot)$ is an HTSAT network (Chen et al., 2022), while the text encoder $E^t(\cdot)$ is a RoBERTa model (Liu, 2019). The HTSAT model takes 10-second audio inputs, and, after an MLP projection layer, it outputs a $L = 32$ frame embedding sequence $\mathbf{e}^{a,loc}(x) \in \mathbb{R}^{L \times d}$. A global audio representation $\mathbf{e}^a(x)$ is obtained by averaging these frame-wise embeddings over time.

We first train an ALM baseline using only the global contrastive objective from §2. This baseline, **FLAM-Global**, is conceptually similar to a CLAP-style model (Wu* et al., 2023), and is trained on the audio-text corpus described in §4, along with Clotho (Drossos et al., 2020) and Audio-Caps (Kim et al., 2019). We train on 10 second audio clips in 48kHz. Further training details can be found in § C.5.

_Table 3._ Zero-shot classification accuracy. FLAM consistently outperforms the baselines across three benchmark datasets, underscoring the advantage of frame-level alignment for robust zero-shot performance. MGA-CLAP* is MGA-CLAP trained on our data.

| Model | ESC50 | US8K | VGGSound |
|---|---|---|---|
| FLAM-Global | 81.6 | 65.4 | 38.9 |
| FLAM | **86.9** | **75.6** | **39.3** |
| MGA-CLAP* | 72.6 | 69.9 | 38.6 |
| LAION-CLAP | 89.1 | 73.2 | 29.1 |
| CompA | 89.1 | 85.7 | 29.5 |
| MGA-CLAP | 94.9 | 83.7 | 31.8 |

Our final system, **FLAM**, initialize from FLAM-Global and trains additionally on a frame-wise contrastive objective $\mathcal{L}_{\text{SED}}$ and a prior loss $\mathcal{L}_{\text{p}}$. The total loss is:

$$\mathcal{L} = \gamma^{\text{CLIP}} \mathcal{L}_{\text{CLIP}} + \gamma^{\text{SED}} \mathcal{L}_{\text{SED}} + \gamma^{\text{p}} \mathcal{L}_{\text{p}},$$

where we set $\gamma^{\text{CLIP}} = 1$, $\gamma^{\text{SED}} = 200$, and $\gamma^{\text{p}} = 1$. We set $\gamma^{\text{CLIP}}$ and $\gamma^{\text{SED}}$ such that $\mathcal{L}_{\text{CLIP}}$ and $\mathcal{L}_{\text{SED}}$ are in same scale, while we found setting $\gamma^{\text{p}} = 1$ is enough to make $\mathcal{L}_{\text{p}}$ converge. This approach allows FLAM to integrate both global sample-level alignment and fine-grained frame-level alignment.

We generate 1 Million mixtures for training using the augmentation procedure where each mixture has a length of 10 seconds. We hold out 5k backgrounds and 10k events, and make 10k mixtures from the held-out events as our primary test set (**Held-out**). For additional evaluation of generalization, we create another 10k test mixtures, **ASFX-SED**, using sound effects from the Adobe Audition SFX Library (ASFX)(Wilkins et al., 2023) that were entirely unseen during training.[2] Since our frame-wise contrastive method accommodates closed-set SED, we train FLAM not only on the synthetic open-vocabulary SED dataset (§4.2) but also on AudioSet-Strong (Hershey et al., 2021), DESED (Serizel et al., 2020), and UrbanSED (Salamon et al., 2017). We retain the global contrastive objective so that FLAM sees the same data as FLAM-Global, thereby reinforcing robust sample-level alignment during the second training stage. Further details of training FLAM is included in C.6.

We compare FLAM to **MGA-CLAP** (Li et al., 2024), which follows a similar architecture (HTSAT + BERT (Devlin, 2018)) and uses only a global contrastive objective. MGA-CLAP adds a shared codebook for text and audio embeddings, adopts a contrastive loss that upweights hard negative samples, and modifies the HTSAT model to be more temporal-aware. We re-train MGA-CLAP on our dataset for direct comparison with FLAM and FLAM-Global.

---

[2]We publicly release the ASFX-SED dataset for future benchmarking: `http://flam-model.github.io/asfx-sed.html`.

## 5.2. SED Performance

Table 1 summarizes results on closed-set datasets (DESED, MAESTRO, AudioSet-Strong, UrbanSED) and two synthetic open-vocabulary test sets (Held-out, ASFX-SED). Acoustic events and audio backgrounds of both Held-out and ASFX-SED datasets are unseen during training. We report PSDS, and MPAUC depending on each dataset's standard metric, and additionally use a frame-based AUROC for all dataset (details of evaluation metrics in § C.7). For open-vocabulary SED datasets, we calculate AUROC by only compute true positive and false positive rates over the events that actually occur in an audio clip.

FLAM achieves substantially better frame-level alignment than both FLAM-Global and MGA-CLAP on nearly every metric. On DESED, MAESTRO, and UrbanSED, FLAM yields improved or comparable AUROCs, while also scoring better PSDS values except DESED dataset. We hypothesize that the PSDS on DESED mainly results from its limited scale, containing only 692 real-world annotated samples, thus could be more prone to higher variance and incomplete coverage of acoustic events. FLAM's performance on the synthetic tasks (Synth, ASFX) is particularly strong, indicating effective generalization to unseen audio events. In contrast, FLAM-Global fares reasonably well on retrieval but underperforms on fine-grained detection, highlighting the need for explicit frame-level training. The MGA-CLAP is indeed better than FLAM-Global on SED metrics, but it performs poorly compared to FLAM, which proves the self-supervised local alignment is less robust than FLAM's direct frame-wise objective. To further validate that FLAM's performance gains result specifically from improved frame-level alignment rather than merely enhanced classification, we introduce an alignment-specific metric based on Spearman correlation, which shows FLAM achieves substantially better temporal alignment compared to baselines (see Appendix C.11).

## 5.3. Accurate and Calibrated Output

FLAM's robust training with logits correction and inference using an unbiased classifier enables it to produce intuitive, calibrated probabilities for event detection. Figure 2 compares FLAM's detection output with MGA-CLAP, with additional examples in § A. FLAM effectively transforms raw cosine frame-text similarity into interpretable probabilities, whereas MGA-CLAP fails to detect certain events and lacks an effective output calibration mechanism.

To demonstrate the effectiveness of our bias-corrected objective, we trained ablated models without per-text logits bias correction and per-text logit scaling. As shown in Figure 3, FLAM consistently achieves higher F1 scores across various thresholds on the ASFX-SED dataset using the same inference formulation. The precision of all models increases

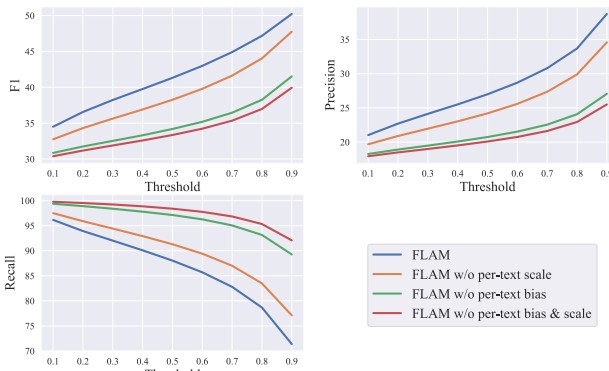

Figure 3. Framewise F1, precision and recall on ASFX-SED dataset. FLAM achieves higher F1 under various threshold compared to ablated models without text-dependent bias and scale, showing the effectiveness of our logit-correction objective.

with the threshold, indicating that FLAM's output is polarized due to the unbiased classifier transforming raw outputs into near-binary probabilities. Additional qualitative results in § B show that FLAM provides more accurately calibrated detection outputs, whereas ablated models produce either overconfident or poorly confident probabilities.

### 5.4. Retrieval Performance

We evaluate text-audio retrieval using recall at ranks 1 and 5 on Clotho (Drossos et al., 2020), AudioCaps (Kim et al., 2019), and the ASFX dataset. Table 2 shows that FLAM and FLAM-Global achieve strong performance on ASFX dataset, indicating that our training corpus generalizes well to the sound effects domain. Existing ALMs like LAION-CLAP and CompA perform better on Clotho and Audio-Caps but only moderately on ASFX, which highlights the limitations of noisy web-scraped audio data those models trained on for specialized sound effects. Additionally, MGA-CLAP trained on our dataset matches FLAM's performance on Clotho and AudioCaps, further demonstrating the significant impact of training distribution. Notably, FLAM's frame-wise training results in minimal degradation of retrieval scores, demonstrating that enhancing local alignment does not compromise sample-level performance.

### 5.5. Zeroshot Classification Performance

Table 3 reports results on ESC-50, US8K, and VGGSound under zero-shot classification, where each dataset's class names serve as the text queries. FLAM surpasses FLAM-Global and MGA-CLAP when all are trained on our corpus, showing that frame-level supervision also refines global representations. Relative to larger-scale ALMs like CompA, FLAM remains competitive, particularly on VGGSound. These results suggest that explicit frame-level alignment enhances overall discriminative ability without sacrificing broad classification performance.

## 6. Related Works

**Sound Event Detection** Sound Event Detection (SED) seeks to detect *which* sound events occur in an audio signal and *where* they occur temporally (Mesaros et al., 2021). Conventional SED systems typically framed as a multi-label, multi-class classification problem over discrete time frames, with a binary cross-entropy (BCE) objective applied at each frame (JiaKai, 2018; Cornell et al., 2024). SED datasets often remain small in both duration and number of classes as temporal annotation of acoustic events is expensive and time consuming. Synthetic methods, e.g., Scaper (Salamon et al., 2017) address label scarcity by mixing isolated sound events with background audio, yielding accurate boundaries at scale. Our approach departs from traditional SED in that we target an *open-vocabulary* setting, where any textual description of a sound event can serve as a query. In contrast to traditional SED data synthesis, our work extends this strategy to an *open-vocabulary* regime, enabling event detection for any textual query.

**Fine-Grained Multi-modal Alignment** Pairing local and global information in the same sample has been successfully explored in the image domain through contrastive representation learning approaches such as Deep Info-Max (Hjelm et al., 2019) and distillation-based methods like DINO (Caron et al., 2021). These techniques enhance performance across general and downstream image tasks. More recently, VLMs including GLIP (Li et al., 2022) have incorporated CLIP-style extensions into object detection pipelines to facilitate zero-shot open-set object detection and grounding. Similarly, Mukhoti et al. (2023) introduce Patch-Aligned Contrastive Learning (PACL) as an unsupervised extension of CLIP for open-vocabulary semantic segmentation. PACL leverages a large-scale captioned image dataset and includes a "variational" CLIP objective that softly classifies image patches through a softmax over their local alignment with the associated caption.

In the audio domain, neither labeled local alignment datasets nor large-scale unsupervised datasets with up to a billion samples are available. Some audio-language models improve temporal reasoning (e.g., "a car crashes before a man cries") by incorporating temporal compositional examples into contrastive objectives, enabling the learning of complex temporal relationships (Ghosh et al., 2024; Wu et al., 2023; Yuan et al., 2024). However, these approaches only achieve instance-level or partial alignment without explicit event localization. Few studies address local text-audio alignment. Text-Audio Grounding (Xu et al., 2021; 2024) aligns word-audio pairs by mapping 527 AudioSet events (Gemmeke et al., 2017) to phrases in the 46k-sample AudioCaps

captions, but it is limited by its ontology and dataset size. MGA-CLAP (Li et al., 2024) represents a more closely related approach that builds on an audio-language contrastive model with a shared codebook to regularize embeddings and trains ALM with a temporal-aware audio encoder using contrastive loss with hard negatives, enhancing temporal correspondence yet still relying on sample-level labels. In contrast, our work leverages ground-truth temporal labels for open-vocabulary, frame-wise SED, integrates synthetic and closed-set data to enhance localization, and enables training on a million-scale supervised dataset.

## 7. Conclusion

In this paper, we introduced FLAM, a frame-wise audio–language modeling framework for open-vocabulary sound event detection. Our approach augments standard audio–text contrastive learning by incorporating a frame-level contrastive objective and a text-dependent logit bias correction mechanism to address severe label imbalance. Through large-scale data synthesis in which we mix labeled sound events into diverse 10-second background clips, we created an extensive training corpus that enabled robust frame-wise supervision. Experimental results demonstrate that FLAM significantly outperforms models trained solely at the clip level in terms of fine-grained event localization, while preserving strong retrieval performance and reliable zero-shot classification. By aligning text-based queries to localized acoustic frames, this work extends the versatility of multimodal audio–language models and opens the door to open-vocabulary sound event detection. The improved temporal precision of FLAM not only benefits open-vocabulary sound event detection but also offers more interpretable global and local representations that could strengthen downstream tasks requiring temporal alignment.

FLAM represents an initial step toward large-scale open-vocabulary sound event detection, but several aspects remain to be improved. The current training corpus, while diverse, is still limited in scale; future work could explore larger and more diverse corpora, potentially by synthesizing additional labeled mixtures or leveraging web-scale audio. The model architecture itself is relatively lightweight, suggesting that scaling up the encoder or introducing more expressive architectures may yield further gains. Additionally, FLAM uses a fixed 10-second audio input and a coarse frame resolution, which constrains its ability to handle longer or more temporally nuanced recordings. Future efforts could focus on supporting variable-length audio and adopting encoders with finer temporal granularity. Beyond architectural and data improvements, future work could explore the use of real-world frame-level annotations, better evaluation protocols, KL penalty to align frame-level outputs with global model, and generative augmentation strategies to further

enhance open-vocabulary localization.

## Impact Statement

This work introduces FLAM, a model for frame-wise audio-language alignment to improve sound event detection using natural language queries. Our goal is to advance the field of multimodal learning by enabling fine-grained and interpretable audio understanding. FLAM may benefit applications such as content indexing, accessibility, and multimedia retrieval. While we do not foresee significant ethical risks, we encourage responsible use of the model in real-world scenarios.

## Acknowledgment

We would like to express our gratitude to the following individuals for their insightful discussions and valuable advice on the project: Yuanbo Hou, Samuel Lavoie.

This work was done while Yusong was interning at Adobe. The contributions of the first three authors are as follows: Yusong Wu was responsible for the model design, code implementation, data pipeline and augmentation, experiment training, and paper writing. Christos Tsirigotis contributed the logit adjustment loss design and the inference classifier, as well as authoring the derivations and proof section in the paper. Ke Chen participated in figure creation, experiment training, model design, and provided technical mentorship throughout the project.

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

# A. Sound Event Detection Results

## A.1. ASFX-SED Dataset

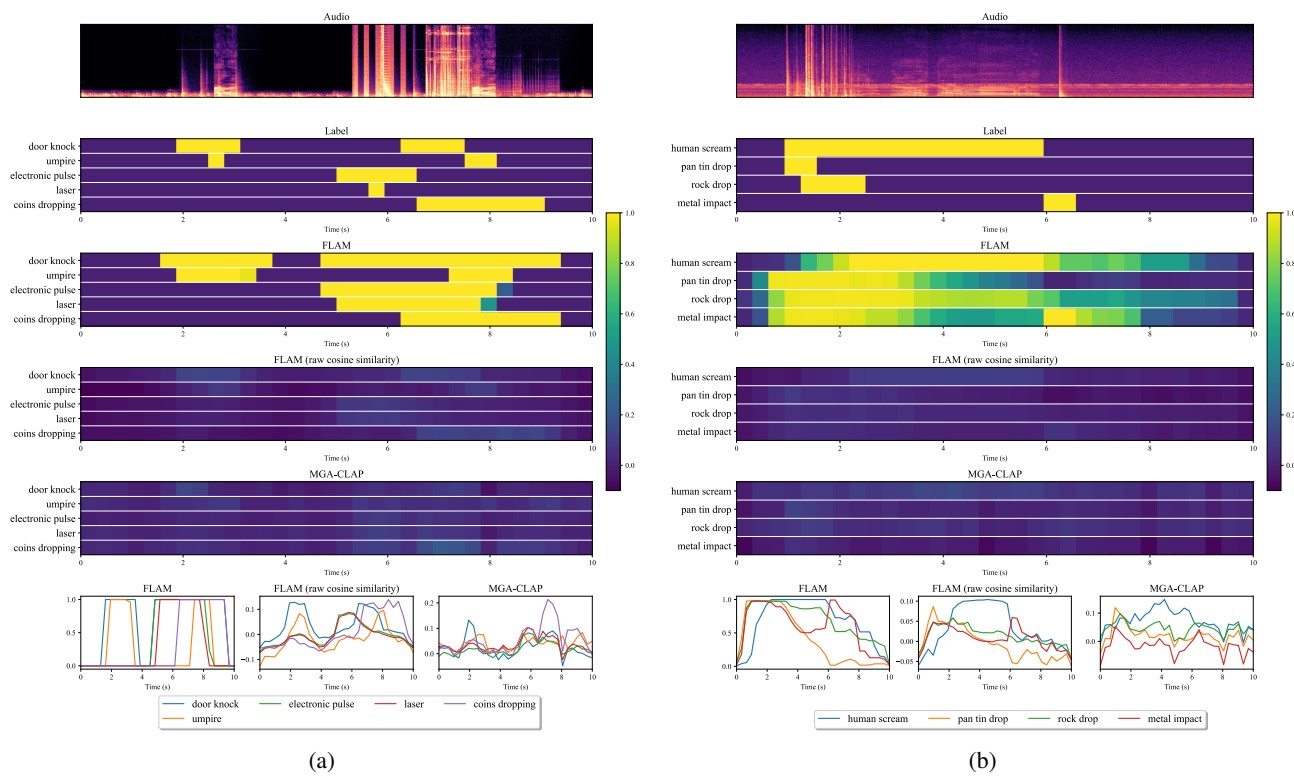

(a)            (b)

*Figure 4.* Sound event detection results of FLAM on ASFX-SED dataset

## A.2. Synthetic Held-out Dataset

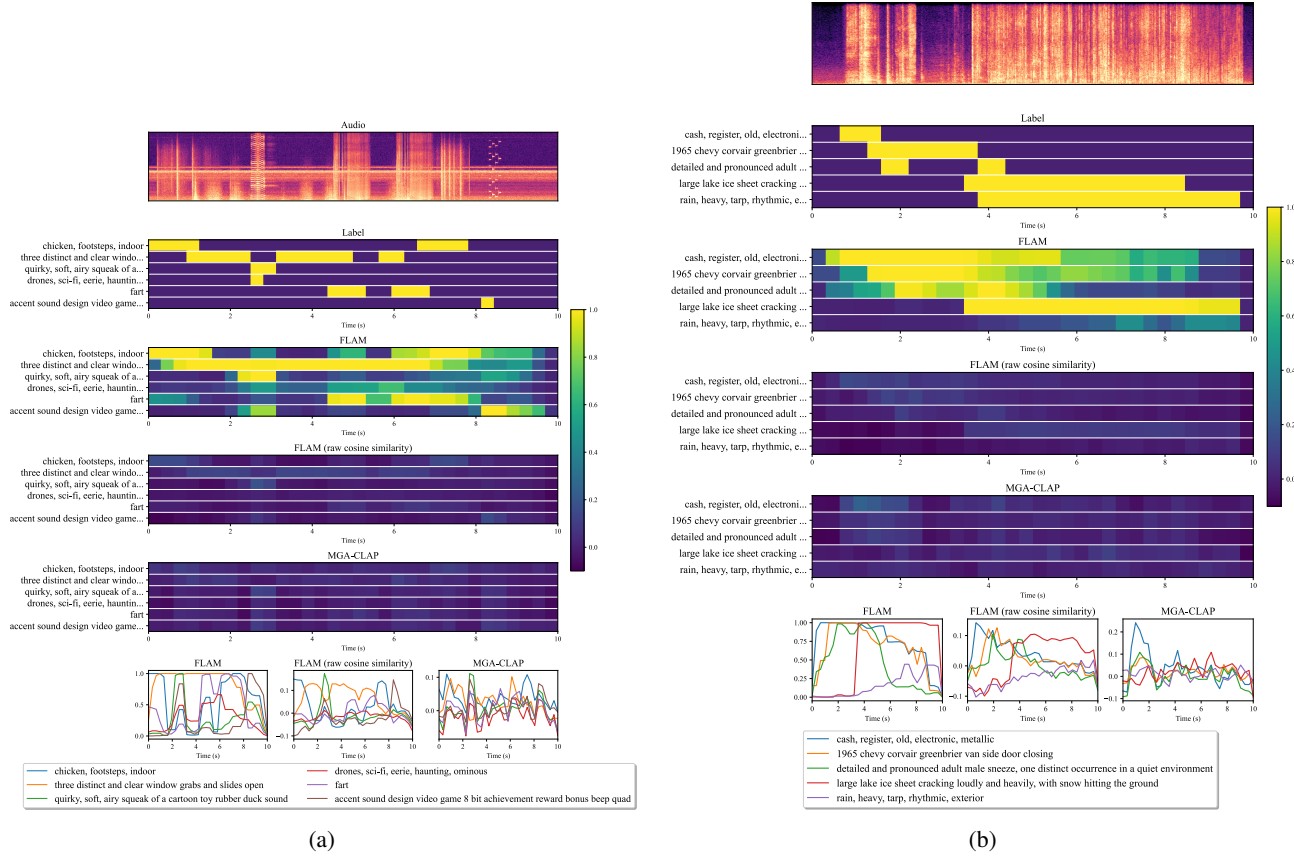

(a)                                                                          (b)

*Figure 5.* Sound event detection results of FLAM on synthetic held-out dataset.

## A.3. Audioset-Strong Dataset

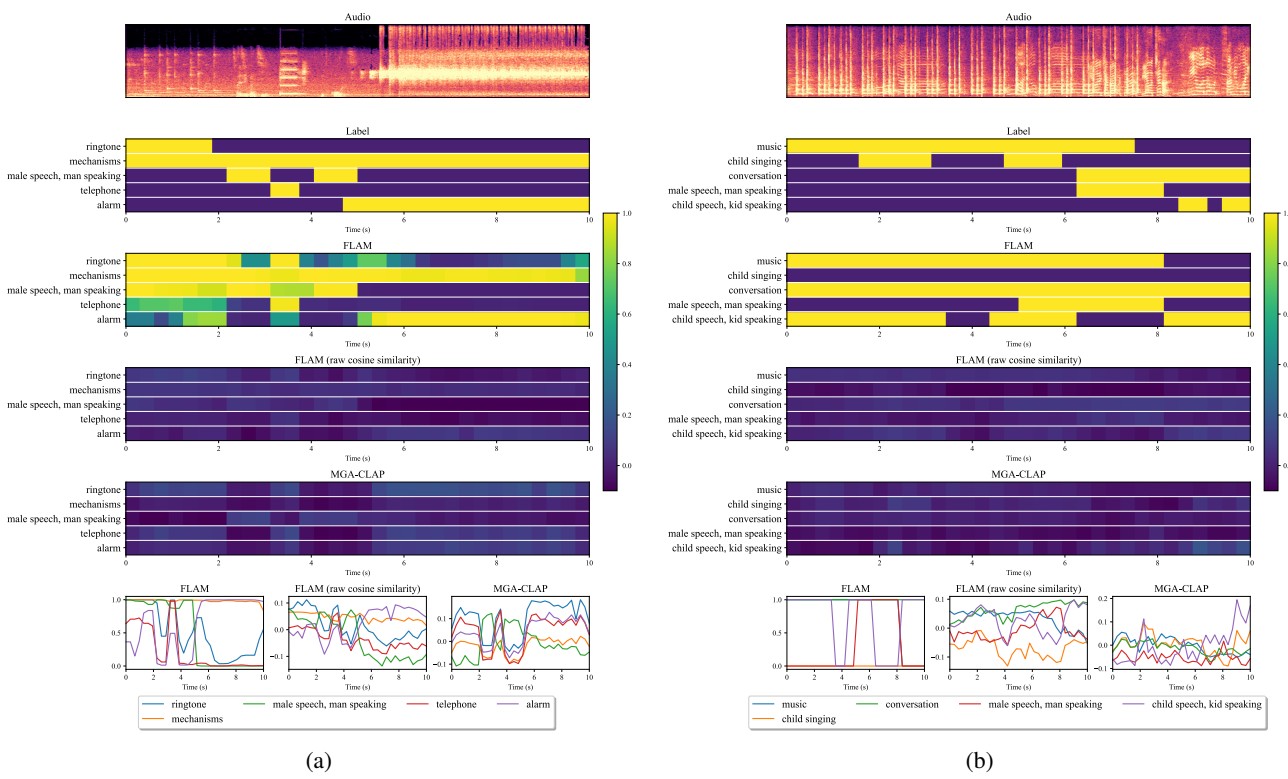

(a)

(b)

*Figure 6.* Sound event detection results of FLAM on Audioset-Strong dataset.

# B. Ablation Models Output Comparison

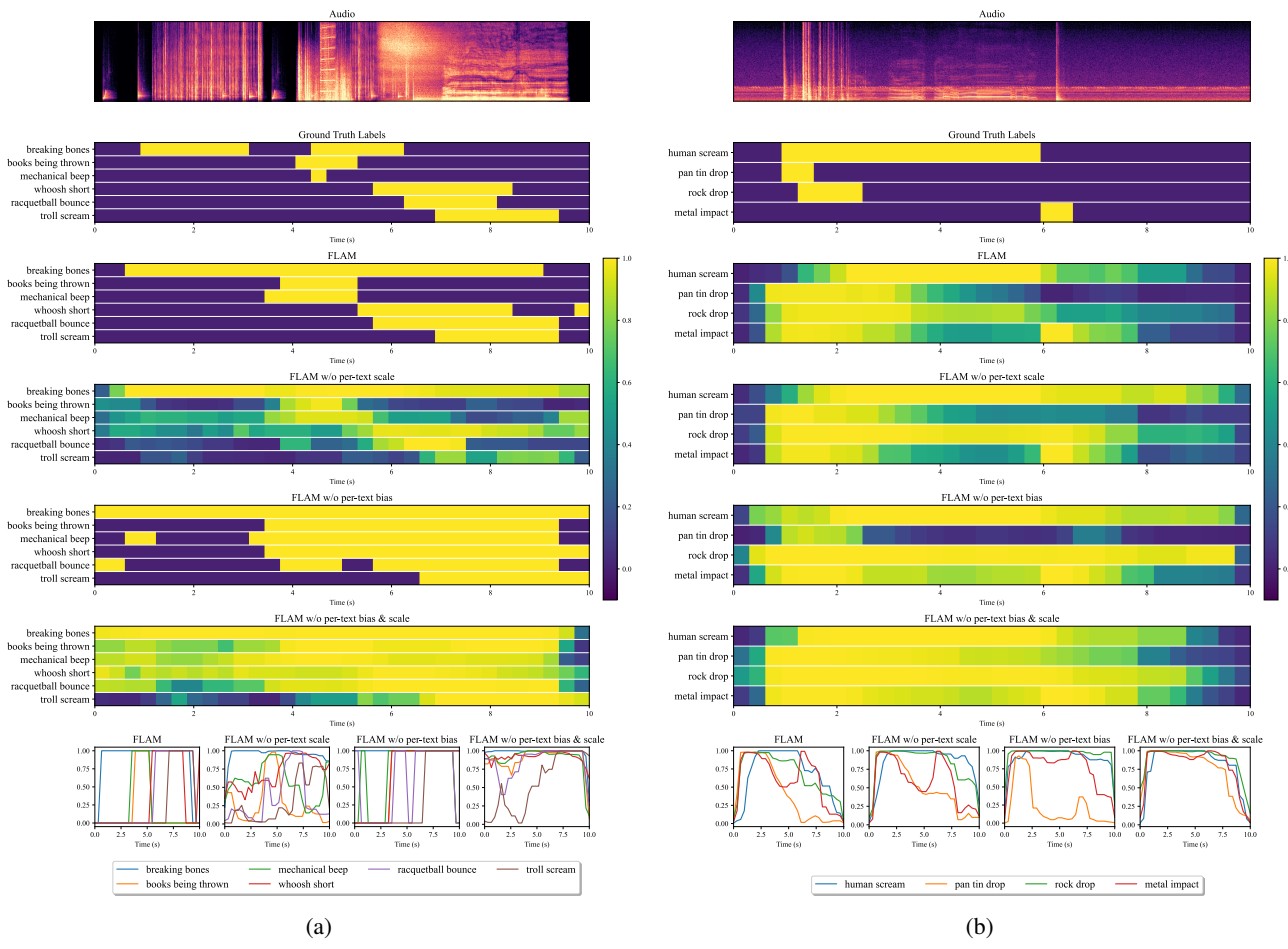

*Figure 7.* Sound event detection results of FLAM on ASFX-SED dataset compared with ablation models.

*Table 4.* Sound event detection performance on synthetic open-vocabulary SED (Held-out, ASFX-SED) and traditional closed-set SED dataset (DESED, MAESTRO, Audioset-strong, UrbanSED) among ablated models.

| Model | Held-out | ASFX-SED | DESED | | MAESTRO | Audioset-S | | UrbanSED | |
|---|---|---|---|---|---|---|---|---|---|
| | AUROC | AUROC | PSDS | AUROC | MPAUC | PSDS | AUROC | PSDS | AUROC |
| FLAM | 91.0 | 81.23 | 9.37 | 91.66 | 56.97 | 11.16 | 94.76 | 29.52 | 93.62 |
| w/o FLAM-Global Init | 91.35 | 80.34 | 6.37 | 89.7 | 52.62 | 10.56 | 94.3 | 29.53 | 92.35 |
| w/o Closed-set SED data | 86.91 | 74.94 | 9.02 | 89.7 | 53.92 | 2.42 | 84.37 | 6.51 | 76.12 |
| w/o Global Loss | 91.33 | 81.15 | 10.25 | 91.98 | 32.45 | 10.77 | 93.89 | 30.68 | 93.08 |

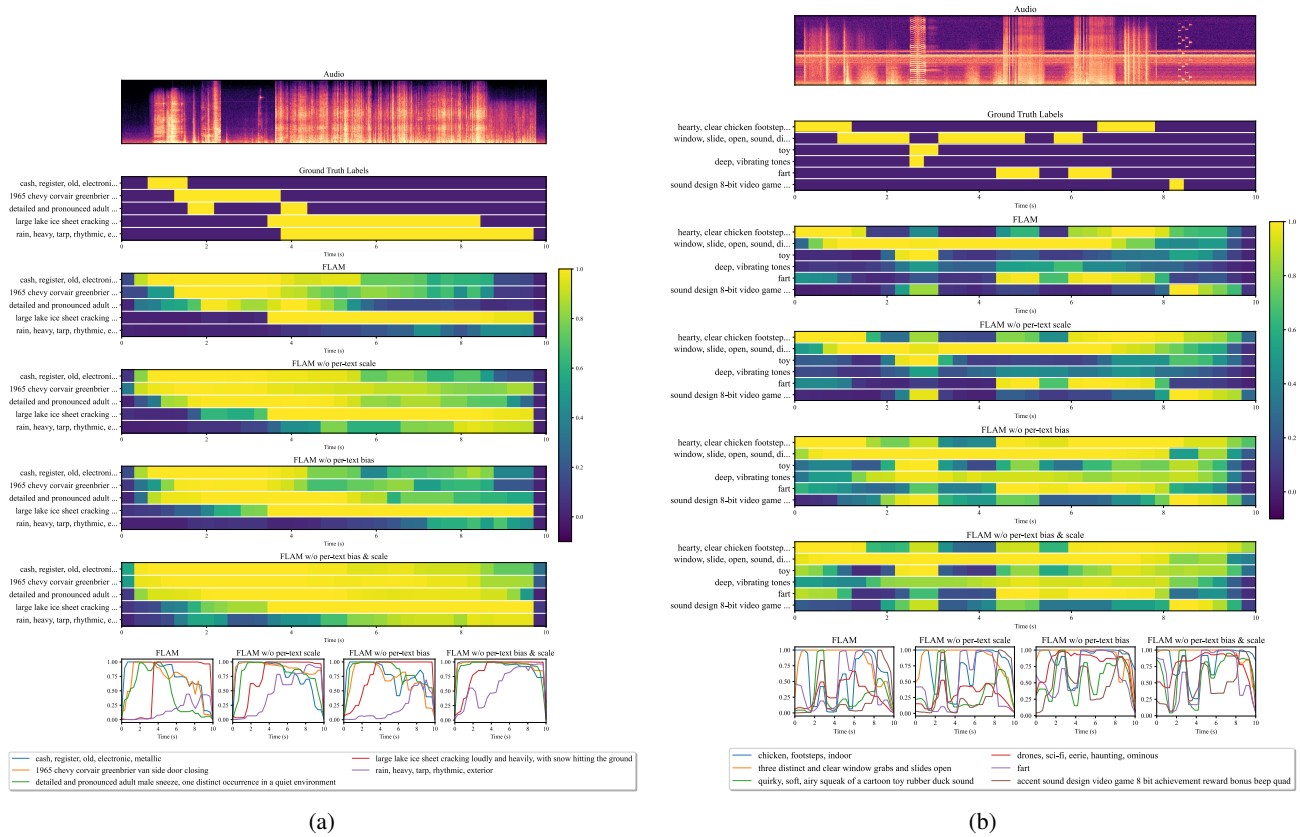

(a)                (b)

*Figure 8.* Sound event detection results of FLAM on synthetic held-out dataset compared with ablation models.

# C. Appendix

## C.1. SigLIP Objective

SigLIP (Zhai et al., 2023) treats contrastive alignment as a *binary classification* problem $\mathcal{L}_{\text{SigLIP}} =$

$$-\frac{1}{B}\sum_{i=1}^{B}\sum_{j=1}^{B}\log\sigma\left(z_{i,j}\left(\alpha\,\mathbf{e}_i^a\cdot\mathbf{e}_j^t+\beta\right)\right),\tag{11}$$

where $z_{i,j}\in\{\pm1\}$ is $+1$ if $A_i$ and $Y_j$ match, and $-1$ otherwise. Here, $\beta$ is a learnable bias, and $\alpha>0$ is again a trainable logit scale factor. This can be viewed as a synthetic binary classification with label $z_{i,j}$ trained with binary cross entropy loss.

In original SigLIP paper, the authors initialize the learnable bias as $-10$ to compensate the fact that most labels are negative labels in the binary classification problem during training. In this paper, we provide a perspective of $\beta$ that is derived from

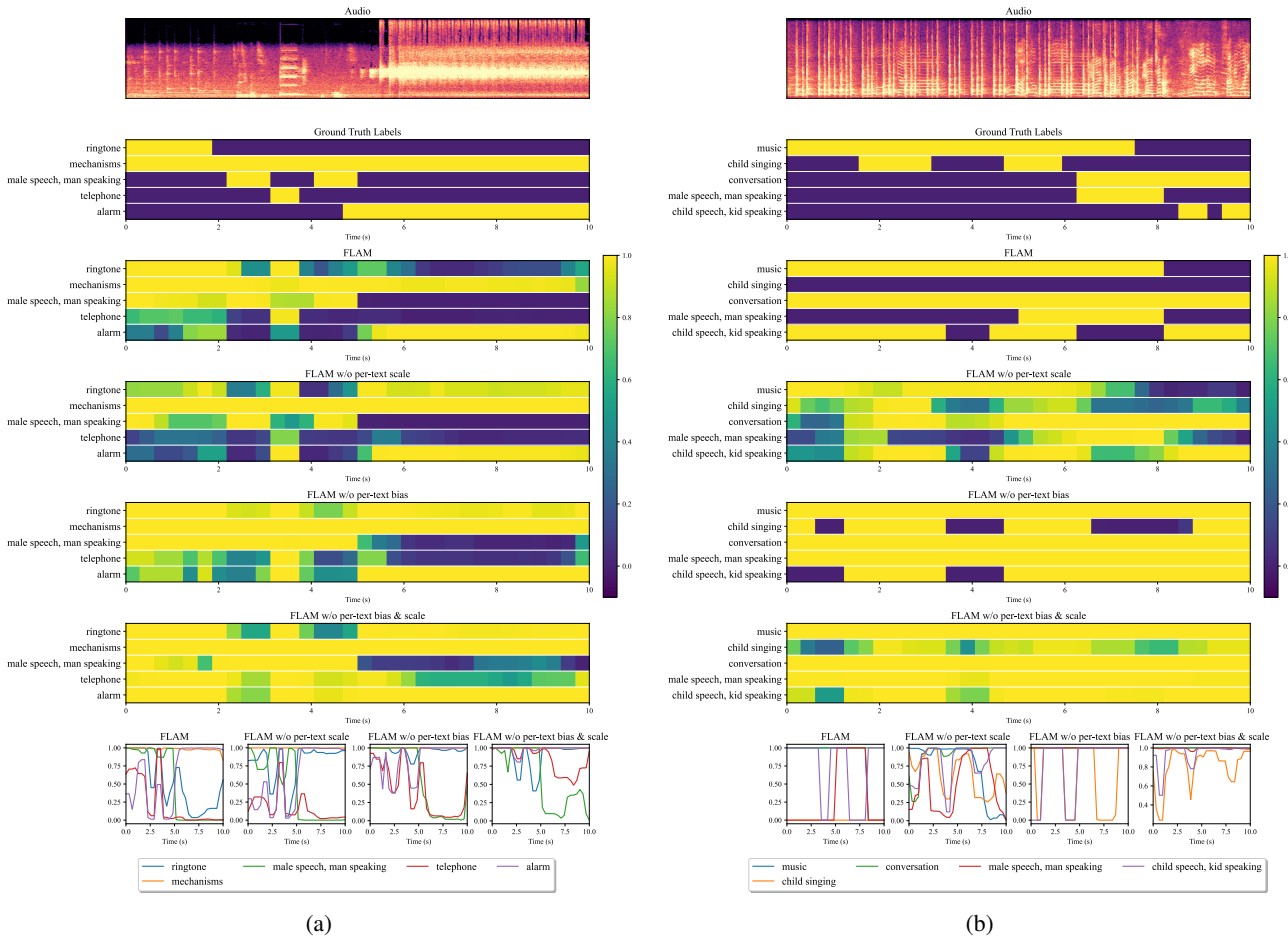

*Figure 9.* Sound event detection results of FLAM on Audioset-Strong dataset compared with ablation models.

the logit adjustment for the label-imbalanced binary classification problem, which ultimately leads to the formulation of the objective in §3.

Consider the binary classification objective in SigLIP objective (Eq. 11) written equivalently as follows, for binary labels $z'_{i,j} = 0.5\,(z_{i,j} + 1) \in \{0,1\}$:

$$-\frac{1}{B} \sum_{i \in [B], j \in [B]} z'_{i,j} \log \sigma\left(h(X_i, Y_j)\right) + (1 - z'_{i,j}) \log \sigma\left(-h(X_i, Y_j)\right), \tag{12}$$

where $\sigma(\cdot)$ is the sigmoid function $\sigma(x) = \frac{1}{1+\exp(-x)}$, and $h(\cdot, \cdot)$ is the logit function, which is defined as:

$$h(x, y) = \alpha\, \mathbf{e}^a(x) \cdot \mathbf{e}^t(y) + \beta. \tag{13}$$

Thus, in expectation and at the minimum, we can view the SigLIP model as learning

$$\sigma\left((h(x, y)\right) \approx p_{\text{data}}(z = 1 \mid x, y). \tag{14}$$

We can see, however, that this conditional distribution is sensitive to the marginal distribution of the labels during training

$p_{\text{data}}(z = 1)$:

$$
\begin{aligned}
p_{\text{data}}(z = 1 \mid x, y) &= \frac{p_{\text{data}}(z = 1, x, y)}{p_{\text{data}}(z = 1, x, y) + p_{\text{data}}(z = -1 \mid x, y)} \\
&= \frac{1}{1 + \frac{p_{\text{data}}(z=-1,x,y)}{p_{\text{data}}(z=1,x,y)}} = \sigma\left(\log \frac{p_{\text{data}}(z = 1, x, y)}{p_{\text{data}}(z = -1, x, y)}\right) \\
&= \sigma\left(\log \frac{p_{\text{data}}(x, y \mid z = 1)}{p_{\text{data}}(x, y \mid z = -1)} + \log \frac{p_{\text{data}}(z = 1)}{p_{\text{data}}(z = -1)}\right) \\
&= \sigma\left(\log \frac{p_{\text{data}}(x, y)}{p_{\text{data}}(x)p_{\text{data}}(y)} + \log \frac{\frac{1}{B}}{\frac{B-1}{B}}\right) \\
&= \sigma\left(\log \frac{p_{\text{data}}(y|x)}{p_{\text{data}}(y)} - \log(B - 1)\right).
\end{aligned}
\tag{15}
$$

The classification problem that SigLIP learns is label-imbalanced due to the outnumbered negative samples compared to positive samples. In the derivation above, we can observe that for very large batch sizes, the term $-\log(B - 1)$ can effectively dominate the logit making $p_{\text{data}}(z = 1 \mid x, y) \to 0$ as $B \to \infty$. If model logit bias $\beta = 0$ in Eq. 13, then the dot product has to learn to be negative for all $x$ and $y$, thus impeding the learning of positive associations. However, by setting it directly so that $\beta = -\log(B - 1)$ we find that

$$
\alpha\, \mathbf{e}^a(x) \cdot \mathbf{e}^t(y) \cancel{+ \beta} = \log \frac{p_{\text{data}}(y|x)}{p_{\text{data}}(y)} \cancel{- \log(B - 1)},
\tag{16}
$$

and the dot product of embeddings is effectively learning the same log-density ratio as CLIP. To verify our reasoning, we note that the optimal batch size and $\beta$ combination in the SigLIP paper was $B = 32768$ and $\beta = -10$. Indeed, the optimal $\beta$ according to our reasoning is $-\log 32768 = -10.397 \approx -10$.

### C.2. Optimum of $\mathcal{L}_{\text{SED}}$

Similarly to the derivation of the model at optimum in expectation for SigLIP in §C.1, we have the following for our proposed loss in Eq. 4

$$
\begin{aligned}
p_{\text{data}}(z = 1 \mid x, l, y) &= \frac{p_{\text{data}}(z = 1, x, l, y)}{p_{\text{data}}(z = 1, x, l, y) + p_{\text{data}}(z = -1 \mid x, l, y)} \\
&= \frac{1}{1 + \frac{p_{\text{data}}(z=-1,x,l,y)}{p_{\text{data}}(z=1,x,l,y)}} = \sigma\left(\log \frac{p_{\text{data}}(z = 1, x, l, y)}{p_{\text{data}}(z = -1 x, l, y)}\right) \\
&= \sigma\left(\log \frac{p_{\text{data}}(x, l \mid y, z = 1)}{p_{\text{data}}(x, l \mid y, z = -1)} + \log \frac{p_{\text{data}}(y, z = 1)}{p_{\text{data}}(y, z = -1)}\right) \\
&= \sigma\left(\log \frac{p_{\text{data}}(x, l \mid y)}{p_{\text{data}}(x, l)} + \log \frac{p_{\text{data}}(z = 1 \mid y)\cancel{p_{\text{data}}(y)}}{p_{\text{data}}(z = -1 \mid y)\cancel{p_{\text{data}}(y)}}\right) \\
&= \sigma\left(\log \frac{p_{\text{data}}(y \mid x, l)}{p_{\text{data}}(y)} + \beta^*(y)\right).
\end{aligned}
\tag{17}
$$

### C.3. Derivation of Robust Classifier

As we have argued in §3.2, we recognize that our proposed $\mathcal{L}_{\text{SED}}$ in Eq. 4 introduces an imbalance in the labels $z \in \{-1, 1\}$ between positive and negative tuples of frames $l$ in audio samples $x$ and event descriptions $y$. Furthermore, we recognise the possibility that our training dataset may impede learning an open-vocabulary detector as it may introduce spurious correlations among the labels $z$ and the contained event description $y$. That is the event "barking" might be more rare in terms of positive frames contained in the dataset, than the even "meowing". However, an open-vocabulary detector should not prioritize learning some events more than others, as we would like it to be able to detect audio frames of relevance regardless of event descriptions provided.

Mathematically, this means that our training and testing conditions of our model may differ. In particular under the training set, $p_{\text{data}}(z = 1 \mid y_1) \neq p_{\text{data}}(z = 1 \mid y_2)$ for different event descriptions $y_1 \neq y_2$. However at test-time, we deploy out

model under a distribution where $p_{\text{test}}(z = 1 \mid y_1) = p_{\text{test}}(z = 1 \mid y_2) = p_{\text{test}}(z = 1)$ for $y_1 \neq y_2$. That is during test-time we assume that label and event description are marginally independent. Moreover, we are equally interested in detecting events, as well as their absence, correctly. This is reflected in a uniform test-time condition across possible labels. In other words, $p_{\text{test}}(z = 1) = U(z = 1) = 0.5$. Finally, we consider that the mechanism, that generates positive or negative frames given a certain event, is invariant between train-time and test-time conditions, $p_{\text{data}}(x, l \mid y, z) = p_{\text{test}}(x, l \mid y, z)$.

Following Menon et al. (2021) and Tsirigotis et al. (2023), the Bayes-optimal robust classifier is given by

$$z^*(x, l, y) = \arg\max_{z \in \{-1, 1\}} p_{\text{test}}(z \mid x, l, y). \tag{18}$$

We use the Bayes rule and the assumed invariance to express an equivalent classifier in terms of the training distribution $p_{\text{data}}$

$$p_{\text{test}}(z \mid x, l, y) \propto_z \frac{p_{\text{test}}(z \mid x, l, y)}{U_{\{-1,1\}}(z)} = \frac{p_{\text{test}}(z \mid x, l, y)}{p_{\text{test}}(z \mid y)} \propto_z p_{\text{test}}(x, l \mid y, z) = p_{\text{data}}(x, l \mid y, z)$$

$$\propto_z \frac{p_{\text{data}}(z \mid x, l, y)}{p_{\text{data}}(z \mid y)}. \tag{19}$$

So, we would like to find a classifier for which

$$z^*(x, l, y) = \arg\max_{z \in \{-1, 1\}} \frac{p_{\text{data}}(z \mid x, l, y)}{p_{\text{data}}(z \mid y)}. \tag{20}$$

In our case, there are two possible outcomes for the classifier $z = 1$ and $z = -1$. Taking the *argmax* involves comparing the two density ratios. Specifically, we decide that $z^*(x, l, y) = 1$ if

$$\frac{p_{\text{data}}(z = 1 \mid x, l, y)}{p_{\text{data}}(z = 1 \mid y)} > \frac{p_{\text{data}}(z = -1 \mid x, l, y)}{p_{\text{data}}(z = -1 \mid y)}$$

$$\frac{p_{\text{data}}(z = 1 \mid x, l, y)}{p_{\text{data}}(z = 1 \mid y)} > \frac{1 - p_{\text{data}}(z = 1 \mid x, l, y)}{1 - p_{\text{data}}(z = 1 \mid y)} \tag{21}$$

$$p_{\text{data}}(z = 1 \mid x, l, y) > p_{\text{data}}(z = 1 \mid y)$$

Essentially, $p_{\text{data}}(z = 1 \mid y)$ determines a threshold for the classifier $p_{\text{data}}(z = 1 \mid x, l, y) = \sigma(h(x, l, y))$ to robustly decide whether frame $(x, l)$ can be described by $y$. Observe that $p_{\text{data}}(z \mid x, l, y)$ has different decision boundary for different prompt $y$. To fix this, we define a new function $s(x, l, y)$ that has unified decision boundary of $0.5$ for all $y$:

$$s(x, l, y) = \frac{p_{\text{data}}(z = 1 \mid x, l, y)}{p_{\text{data}}(z = 1 \mid x, l, y) + p_{\text{data}}(z = 1 \mid y)}. \tag{22}$$

Notice that $s(x, l, y) > 0.5 \Leftrightarrow p_{\text{data}}(z = 1 \mid x, y) > p_{\text{data}}(z = 1 \mid y)$.

In practice, we can use $\sigma\left(\frac{p_{\text{data}}(y|x,l)}{p_{\text{data}}(y)}\right)$ to get approximate value of $s(x, l, y)$ when $\beta^*(y)$ is negative enough for all $y$. This is because:

$$s(x, l, y) = \frac{p_{\text{data}}(z = 1 \mid x, l, y)}{p_{\text{data}}(z = 1 \mid x, y) + p_{\text{data}}(z = 1 \mid y)}$$

$$= \frac{1}{1 + \frac{p_{\text{data}}(z=1|y)}{p_{\text{data}}(z=1|x,l,y)}} = \frac{1}{1 + \frac{\sigma(\beta^*(y))}{\sigma\left(\log \frac{p_{\text{data}}(y|x,l)}{p_{\text{data}}(y)} + \beta^*(y)\right)}}$$

$$= \frac{1}{1 + \frac{1 + e^{-(\log \frac{p_{\text{data}}(y|x,l)}{p_{\text{data}}(y)} + \beta^*(y))}}{1 + e^{-\beta^*(y)}}} \rightarrow \frac{1}{1 + \frac{e^{-\log \frac{p_{\text{data}}(y|x,l)}{p_{\text{data}}(y)}}(-e^{-\beta^*(y)})}{-e^{-\beta^*(y)}}} \tag{23}$$

$$= \sigma\left(\log \frac{p_{\text{data}}(y \mid x, l)}{p_{\text{data}}(y)}\right) \quad \text{as} \quad \beta^* \rightarrow -\infty.$$

In practice, we observe $\beta^*$ to be near $-8$, which result in neglectble difference before and after the approximation. By approximating $s(x, l, y)$ with $\sigma\left(\log \frac{p_{\text{data}}(y|x,l)}{p_{\text{data}}(y)}\right)$, we save the compute for estimating $\beta^*(y)$ during inference.

### C.4. Loudness Relabel

We compute the RMS curve using a window size of 2400 and a hop size of 1200 (50 Hz at a 48 kHz sample rate). Post-processing proceeds as follows: (1) remove negative segments shorter than 10 frames (200 ms) if they lie between positive segments (mark them as positive), (2) remove positive segments shorter than 2 frames (40 ms) if the total event exceeds 10 frames (200 ms).

### C.5. Training Details for FLAM-Global

Following LAION-CLAP (Wu* et al., 2023), we initialize HTSAT and RoBERTa from pretrained checkpoints. We use a batch size of 768, a learning rate of $10^{-4}$, and an Adam optimizer with $\beta_1 = 0.9$, $\beta_2 = 0.99$. The learning rate schedule employs cosine warmup (3200 steps) followed by linear decay, for a total of 50,000 steps.

All sampled captions are converted to lower case, with a maximum text input length of 77 tokens. We sample data from our (1.1M) dataset, AudioCaps, and Clotho at weights of (1.0, 0.1, 0.1). During training, 30% of the time we randomly downsample audio to 16 kHz or 32 kHz, then upsample back to 48 kHz. For each audio, we randomly choose one of its captions and one of its tags, forming the training text as "`keyword, tag`". We also removed all samples from evaluation datasets from the general audio training data.

### C.6. Training Details for FLAM

We train FLAM with a batch size of 512 and a learning rate of $10^{-4}$, using Adam ($\beta_1 = 0.9$, $\beta_2 = 0.99$) and the same warmup-then-decay schedule with 3200 steps of warmup and train 120,000 steps. When sampling, we use our dataset, AudioCaps, and Clotho with weights (1.0, 0.1, 0.1). Additionally, we sample from synthetic SED data, AudioSet-strong, UrbanSED, and DESED with weights (0.5, 0.5, 0.1, 0.1). We again apply the same audio downsampling strategy as in FLAM-Global.

To improve event-caption generalization, we replace LLM-generated captions with random tags 50% of the time. We initialize the last bias layer of the per-text bias MLP to -8 (matching the average converged logit bias in $\mathcal{L}_p$) and the last bias layer of the per-text scale MLP to $\log(10)$, following CLIP (Radford et al., 2021) and SigLip (Zhai et al., 2023).

### C.7. Details of SED Metrics

Following MGA-CLAP (Li et al., 2024), we apply a median filter of size 3 frames to the SED output before evaluation. We use `sed_scores_eval` (Ebbers et al., 2022) to compute the following:

- **PSDS**, (or refereed to as "PSDS1" in other works) with DTC=0.7, GTC=0.7, $\alpha_{ST} = 1$, $\alpha_{CT} = 0$, and $e_{max} = 100$.

- **MPAUC**, the mean partial AUC on segment-level ROC curves (1 s segments), capped at an FPR of 0.1.

- **AUROC**, the full ROC AUC on segments of length 0.3125 s (the HTSAT frame length).

### C.8. Training Details of MGA-CLAP

We follow MGA-CLAP (Li et al., 2024) in hyperparameters and architectures except that: (1) we use 77 input text tokens instead of 30, (2) we train for 50,000 steps to match FLAM-Global.

### C.9. Details of Ablation Models

The ablated models are defined as follows:

- **FLAM without per-text scale** uses a scalar scale initialized to $\log(10)$, which is updated by the SED loss.

- **FLAM without per-text bias** employs a scalar bias initialized to $-10$, following SigLIP's default, and this bias is updated by the SED loss.

- **FLAM without per-text bias and scale** incorporates both a scalar scale initialized to $\log(10)$ and a scalar bias initialized to $-10$.

All ablated models utilize the same hyperparameters as the original FLAM model.

### C.10. Ablations on SED Training

We conducted ablation experiments to investigate the impact of three factors: (1) removing FLAM-Global initialization (i.e., initialization from pre-trained HTSAT and RoBERTa), (2) removing the global loss during SED training, and (3) removing the closed-set SED dataset. Table 4 presents the results. While removing FLAM-Global initialization and global loss leads to only a minor drop in performance, removing the closed-set SED dataset causes a significant performance degradation, underscoring the importance of closed-set data in our SED training.

### C.11. Alignment Correlation Metric

To robustly quantify the alignment quality between audio frames and textual descriptions, we propose an alignment-specific metric based on Spearman's rank correlation coefficient ($\rho$). This metric directly measures how well the model's frame-level similarity predictions correspond to the actual temporal occurrence of audio events, independent of absolute similarity magnitudes or decision thresholds.

Formally, for each captioned event, we compute Spearman correlation between:

- The model's predicted output for each frames.

- The binary ground-truth event presence labels for corresponding frames.

A higher Spearman $\rho$ indicates more accurate temporal alignment of predictions.

Table 5 shows FLAM significantly outperforms baseline models (MGA-CLAP, FLAM-Global) on two open-vocabulary datasets (**ASFX-SED** and synthetic **Held-out**), confirming that the improved SED performance of FLAM indeed arises from enhanced frame-level alignment.

*Table 5.* Spearman rank correlation ($\rho$) measuring alignment quality. FLAM achieves significantly higher alignment correlations than baseline methods on both datasets, indicating superior temporal alignment. All reported correlations have a p-value $< 0.01$.

| Model | ASFX-SED $\rho$ | Held-out $\rho$ |
|---|---|---|
| FLAM | **0.409** | **0.600** |
| MGA-CLAP | 0.256 | 0.352 |
| FLAM-Global | 0.197 | 0.262 |

### C.12. Additional Ablation Experiments

We present further ablation experiments to clarify the impact of the global loss component and temporal granularity (number of frames, $L$).

**Global Loss Ablation**    Removing the global contrastive loss (FLAM - no global loss) resulted in a slight improvement in SED metrics but a significant drop in retrieval and zero-shot classification performance. As shown in Tables 6, 7, and 8, the global loss is essential to maintain robust global alignment required for effective retrieval tasks. This indicates a critical trade-off between local alignment objectives for SED and global representation quality necessary for retrieval and classification.

**Temporal Granularity Ablation** ($L = 128$)    Increasing temporal resolution from $L = 32$ to $L = 128$ (FLAM - $L = 128$) yielded minor gains in SED performance but adversely affected retrieval and zero-shot classification accuracy, suggesting that finer temporal resolution may encourage overly specialized frame-level embeddings at the expense of globally coherent audio representations.

The detailed ablation results are summarized in Tables 6, 7, and 8.

These additional experiments reinforce the necessity of joint global-local training, highlighting trade-offs inherent in temporal granularity choices, and underscore the importance of balanced frame-level and global objectives.

*Table 6.* Sound event detection performance on synthetic open-vocabulary SED (Held-out, ASFX-SED) and traditional closed-set SED dataset (DESED, MAESTRO, Audioset-strong, UrbanSED).

| Model | Held-out AUROC | ASFX-SED AUROC | DESED PSDS | DESED AUROC | MAESTRO MPAUC | Audioset-S PSDS | Audioset-S AUROC | UrbanSED PSDS | UrbanSED AUROC |
|---|---|---|---|---|---|---|---|---|---|
| FLAM-Global | 67.76 | 65.14 | 7.09 | 85.52 | 51.13 | 1.11 | 82.54 | 0.82 | 67.39 |
| FLAM (proposed) | **91.0** | **81.23** | 9.37 | **91.66** | **56.97** | **11.16** | **94.76** | **29.52** | **93.62** |
| MGA-CLAP* | 74.17 | 69.56 | **14.72** | 89.28 | 52.50 | 1.24 | 79.12 | 6.42 | 78.22 |
| FLAM - no global loss | 91.33 | 81.15 | 10.25 | 91.98 | 32.45 | 10.77 | 93.89 | 30.68 | 93.08 |
| FLAM - $L = 128$ | 92.6 | 82.51 | 9.52 | 92.45 | 49.79 | 11.3 | 94.83 | 36.0 | 95.1 |
| MGA-CLAP (reported) | - | - | **26.4** | - | - | 10.1 | - | 8.7 | - |

*Table 7.* Recall performance of text to audio (T2A) and audio to text (A2T) retrieval.

| Model | Dataset | ASFX T2A R@1 | ASFX T2A R@5 | ASFX A2T R@1 | ASFX A2T R@5 | Clotho T2A R@1 | Clotho T2A R@5 | Clotho A2T R@1 | Clotho A2T R@5 | AudioCaps T2A R@1 | AudioCaps T2A R@5 | AudioCaps A2T R@1 | AudioCaps A2T R@5 |
|---|---|---|---|---|---|---|---|---|---|---|---|---|---|
| FLAM - Global | | **4.4** | 14.8 | **4.0** | 13.8 | **14.3** | **35.8** | 17.9 | 39.8 | 36.0 | **70.5** | 46.1 | **78.6** |
| FLAM | FLAM-Collection | **4.4** | **15.2** | 3.9 | **13.9** | 13.8 | 33.2 | 16.7 | **42.2** | 32.1 | 64.8 | 43.3 | 75.0 |
| MGA-CLAP* | | 3.9 | 14.8 | 3.9 | 13.8 | 13.4 | 30.3 | **18.7** | 39.1 | **36.7** | 69.9 | **47.2** | 78.3 |
| FLAM - no global loss | FLAM-Collection | 0.8 | 2.8 | 1.4 | 5.0 | 5.4 | 13.5 | 8.1 | 23.0 | 3.9 | 20.3 | 7.1 | 23.6 |
| FLAM - $L = 128$ | | 4.4 | 14.8 | 3.8 | 13.4 | 11.6 | 29.4 | 16.0 | 38.0 | 21.8 | 53.1 | 28.9 | 64.0 |
| **Reported Performance from Prior Studies (but trained on different datasets)** | | | | | | | | | | | | | |
| LAION-CLAP | LAION-Audio-630K | 2.0 | 7.6 | 1.6 | 6.0 | 16.1 | 38.3 | 22.7 | 48.5 | 36.1 | 71.8 | 46.8 | 82.9 |
| CompA | CompA-Collection | - | - | - | - | 16.8 | 43.5 | 23.9 | 50.7 | 36.1 | 78.6 | 47.8 | 83.5 |
| MGA-CLAP | WavCaps | 2.3 | 8.3 | 2.0 | 7.4 | 20.4 | 46.0 | 25.3 | 51.2 | 41.8 | 76.1 | 54.4 | 83.6 |

*Table 8.* Zero-shot classification accuracy.

| Model | ESC50 | US8K | VGGSound |
|---|---|---|---|
| FLAM-Global | 81.6 | 65.4 | 38.9 |
| FLAM | **86.9** | **75.6** | **39.3** |
| MGA-CLAP* | 72.6 | 69.9 | 38.6 |
| FLAM - no global loss | 66.9 | 70.4 | 18.5 |
| FLAM - $L = 128$ | 83.1 | 79.0 | 33.3 |
| LAION-CLAP | 89.1 | 73.2 | 29.1 |
| CompA | 89.1 | 85.7 | 29.5 |
| MGA-CLAP | 94.9 | 83.7 | 31.8 |

