# OpenReview forum: "FLAM: Frame-Wise Language-Audio Modeling"
_ICML.cc/2025/Conference — ICML 2025 poster_

### Official Review · Reviewer_Rbvm · 2025-03-12

**Overall Recommendation:** 4

**Summary:**

The paper develops a contrastive audio-language model that is capable of doing frame-level sound event detection. The paper's focus is on using contrastive techniques and off-the-shelf encoders, as well as the correction of bias caused by the imbalance in event labels. The contrastive training, logit adjustment for bias correction and the SED task are all trained for simultaneously, and show good performance compared to recent models in terms of SED, while showing extremely accurate frame alignment of events. The main contribution of the paper is the focus on frame-level SED and the model is trained on a combination of synthetically constructed data (by using sound effect and general audio).

**Claims And Evidence:**

1. It is claimed that the negative examples in contrastive learning causes models to be biased - derivations and prior literature confirm this claim.
2. The authors claim that frame-level SED is superior to instance-level SED - they somewhat demonstrate this by retraining MGA-CLIP on their data, as well as via FLAM-GLOBAL. This does not become generally true unless this method scales up and can outperform state-of-the-art models, but for the scale discussed in the paper, is sufficient.
3. It is claimed that the method leads to highly accurate alignment of SED outputs - this is only demonstrated by example, not via summary statistics or dataset-level measures

**Essential References Not Discussed:**

No major concerns

**Experimental Designs Or Analyses:**

Yes, experiments are appropriately designed, with comparisons to baselines trained on the same data as well as different data. The authors conduct experiments both on held-out test sets of the training data and zero-shot classification studies, and as an added task, also study retrieval. The experiments are fair, with no clear advantages given to the proposed method in comparison to the baseline.

**Methods And Evaluation Criteria:**

The authors measure performance on three appropriate tasks - synthetic open vocab/closed SED, Text-to-audio and Audio-to-text retrieval, as well as zero-shot SED.

The methods used are standard and appropriate, and combining frame-level approaches with instance-level ones is a plausible thing to do in the audio domain.

**Other Comments Or Suggestions:**

None

**Other Strengths And Weaknesses:**

Strengths:
1. The method is simple and elegant, and seems to perform competitively to standard methods from the literature.
2. The focus on bias correction and robustness is admirable and should be standard in the contrastive literature.
3. The synthetic dataset construction is well-justified and the methods to create this data are appropriately explained.

Weaknesses:
1. The improvment in alignment of the audio events, the whole point of doing frame level SED, is only shown through examples, and not robustly checked and justified. This is the paper's major weakness, because, without summary statistics and robust justification, it is not clear if the models improvements in accuracy come from better alignment or simply better classification accuracy. While it is probably that the improvements come from better alignment, I'd like to see this measured better than via examples and diagrams.

**Questions For Authors:**

1. Could the authors come up with a metric that focuses purely on alignment and compare the various models on it? If the frame level approach were strongly justified, I would increase my rating.

**Relation To Broader Scientific Literature:**

The key contributions of the paper build off of the broader scientific literature in the world of sound event detection, with the authors using well-known models such as HTSAT [Chen et. al., 2022] and RoBERTa [Liu, 2019] to develop their methods. The authors also engage with the literature by comparing to standard baselines like MGA-CLAP [Li, 2024] on well-known datasets such as AudioSet

**Theoretical Claims:**

I checked the theoretical claims around bias correction at a high level, and am broadly satisfied with them. I also reviewed the appendix and broadly found the theoretical claims to be correct.

---

> ### Author Rebuttal · Authors · 2025-03-31
>
> We sincerely thank Reviewer Rbvm for their thoughtful and constructive review. We are especially grateful for your recognition of our theoretical and mathematical contributions. As you noted, “the theoretical claims around bias correction… [are] broadly correct,” and your review confirmed that you “reviewed the appendix and broadly found the theoretical claims to be correct.”
>
> We also appreciate your positive assessment of our methodological choices—“the method is simple and elegant”—as well as your acknowledgment that our bias correction and synthetic dataset construction are “well-justified.” Your remark that “the focus on bias correction and robustness is admirable and should be standard in the contrastive literature” particularly resonates with our goals. We are hopeful that our logit-adjusted contrastive loss formulation will inspire future work in other domains such as vision and language.
>
> Below, we address your primary concern and introduce a new alignment-specific evaluation metric, as requested.
>
> ---
> > The improvement in alignment of the audio events, the whole point of doing frame level SED, is only shown through examples, and not robustly checked and justified. This is the paper's major weakness, because, without summary statistics and robust justification, it is not clear if the models improvements in accuracy come from better alignment or simply better classification accuracy. While it is probably that the improvements come from better alignment, I'd like to see this measured better than via examples and diagrams.
> > Could the authors come up with a metric that focuses purely on alignment and compare the various models on it?
>
>
> We fully agree with the reviewer that robust evaluation of alignment quality is essential to justifying the benefits of frame-level supervision. In our original submission, we report frame-level AUROC and PSDS, both of which reflect the model’s ability to distinguish when an event occurs, given a caption. AUROC is computed over all frame-caption pairs, including negative ones, and thus directly reflects alignment fidelity. PSDS (Polyphonic Sound Detection Score) is a DCASE challenge metric that integrates precision and recall across thresholds with a focus on time-sensitive detection.
>
> In Table 1 of the main paper, we showed that FLAM significantly improves both AUROC and PSDS across open-vocabulary and closed-set datasets, which we argued reflects improved temporal alignment. However, we agree that these classification-based metrics only measure alignment indirectly.
>
> To address this, we propose a new diagnostic metric specifically designed to directly evaluate alignment: **Spearman correlation between the model’s frame-text similarity scores and the ground truth label mask**. This measures how well the model’s similarity predictions track the actual occurrence of audio events over time, independent of decision thresholds or absolute similarity magnitude.
>
> **Definition**: For each captioned event, we compute the Spearman rank correlation between:
> - the model’s similarity scores across frames (either pre-sigmoid logits or dot products), and
> - the corresponding binary ground-truth label mask indicating event presence over time.
>
> A higher Spearman correlation indicates that the model more faithfully aligns audio with text on a frame level.
>
> ---
>
> #### **Results: Alignment Correlation (Spearman ρ)**
>
> | Model        | ASFX ρ  | Synth ρ |
> |--------------|---------|---------|
> | **FLAM**         | **0.409** | **0.600** |
> | MGA-CLAP     | 0.256   | 0.352   |
> | FLAM-Global  | 0.197   | 0.262   |
>
> Note: for all Spearman ρ, the p-value is smaller than 0.01.
>
> ---
>
> These results confirm that **FLAM achieves substantially higher alignment correlation** than both MGA-CLAP and FLAM-Global on both ASFX-SED and synthetic datasets. This supports our hypothesis that FLAM’s improvements stem from better temporal alignment, not merely improved classification.
>
> We will consider including this new metric, its formulation, and the results above in the revised manuscript. Thank you again for your insightful suggestion—it has helped strengthen the empirical justification for our frame-level approach.

---

> > ### Comment · Reviewer_Rbvm · 2025-04-02
> >
> > Thank you for adding this metric and justification, I have updated my score accordingly.

---

> > > ### Author Response · Authors · 2025-04-04
> > >
> > > Thank you very much for taking the time to reconsider our response and for updating your score accordingly! We greatly appreciate your constructive feedback and valuable suggestions, which have significantly improved the quality of our paper.

---

### Official Review · Reviewer_GGdQ · 2025-03-13

**Overall Recommendation:** 4

**Summary:**

The paper introduces an open vocabulary SED model, FLAM, trained with sigmoid loss. It outperforms the baseline MGA-CLAP on open vocabulary SED datasets and most closed-set SED datasets. The model is also tested in other tasks such as retreival and classification.

**Claims And Evidence:**

Exisiting contrastively trained multimodal models are not accurate in providing frame-level results. FLAM extends CLAP to frame-level granuarity. To mitigate increased computational cost and biased class variations in available audio-text datasets, sigmoid loss, proposed in SigLIP, is introduced.

**Essential References Not Discussed:**

The reference section covers a descent amount of past litterature

**Experimental Designs Or Analyses:**

It would be insightful to see a comparison in closed-set SED tasks between other SED methods not designed for open vocabulary SED. Ablation studies to confirm the strength of sigmoid loss, its low computational cost and its insensitivity to batch size, should be performed under the SED scenario

**Methods And Evaluation Criteria:**

The proposed architecture and training scheme make sense. The datasts and metrics used in the experiments are standard in this field

**Other Comments Or Suggestions:**

- In table 2, for the A2T experiment on Clotho, should MGA-CLAP be made in bold?
- Some typos?

Sec. 3.3 "we allocate up t slots perup to five times the audio in a batchd pad any unused sloaudio or text ts with placeholders"

Sec. 5 "How does FLAM fare on downstream tasks typically used to evaluate contrastive ALMs?"

Sec. 5.2 "ustic"

**Other Strengths And Weaknesses:**

[Strength]
- FLAM mostly outperforms baselines in some audio-text tasks
- First SED system trained with sigmoid loss in a contrastive learning setting
- Proprietary high-quality sound dataset

[Weakness]
- The strength of sigmoid loss in SED tasks is not verified in the experiments nor discussed in the paper
- FLAM sometimes underperforms MGA-CLAP in audio-text retreival tasks
- Some typos remain

**Questions For Authors:**

Please see "Experimental Designs Or Analyses" and let me know your thoughts

**Relation To Broader Scientific Literature:**

Different from othre audio fields such as sound separation, open vocabulary SED is relatively unexplored field

**Theoretical Claims:**

Sigmoid loss can reduce computational and memory costs because it does not need to compute the global view necessary in InfoNCE loss. It is also evident from the equations of InfoNCE and sigmoid loss that the latter is not affected by batch size

---

> ### Author Rebuttal · Authors · 2025-03-31
>
> We sincerely thank Reviewer **GGdQ** for their thoughtful and constructive feedback, and for recognizing several key strengths and contributions of our work. We are encouraged by your positive assessment of our model and contributions—for instance, your remark that "FLAM mostly outperforms baselines in some audio-text tasks", your recognition that FLAM is the "first SED system trained with sigmoid loss in a contrastive learning setting", and your appreciation of our "proprietary high-quality sound dataset". We also value your comment that "the proposed architecture and training scheme make sense", and your confirmation that "the datasets and metrics used in the experiments are standard in this field". We found your comments particularly insightful and have revised the manuscript to incorporate your suggestions, which we address point-by-point below.
>
> ---
>
> > It would be insightful to see a comparison in closed-set SED tasks between other SED methods not designed for open vocabulary SED.
>
> We appreciate your suggestion to compare FLAM against strong closed-set SED models. In response, we conducted additional evaluations of FLAM on closed-set SED benchmarks alongside state-of-the-art closed-set systems. Due to space constraints, we summarize these results here: https://flam-model.github.io/response_reviewer_ggdq.html
>
> While FLAM performs competitively, especially considering its generalization capabilities, there remains a performance gap compared to highly specialized closed-set models. This result is expected, as closed-set SED methods are explicitly trained on a fixed, predefined label set and can thus heavily optimize for those specific categories. However, this design inherently limits their applicability to new or unseen sound events, since they cannot recognize or adapt to events outside their training vocabulary. In contrast, FLAM supports open-vocabulary detection by leveraging natural language descriptions, enabling broader generalization and better alignment with real-world, evolving sound taxonomies.
>
> From a downstream application perspective, this flexibility makes FLAM more broadly applicable, particularly in dynamic or user-driven scenarios where defining a fixed vocabulary in advance is infeasible. We believe these findings motivate further research into bridging the gap between open and closed-set SED and enhancing generalization across domains.
>
> ---
>
> > Ablation studies to confirm the strength of sigmoid loss, its low computational cost and its insensitivity to batch size, should be performed under the SED scenario
>
> Thank you for encouraging a deeper investigation into the role of sigmoid loss. We conducted the following ablation studies to evaluate its effectiveness under the SED setting, with results in https://flam-model.github.io/response_reviewer_ggdq.htm
>
> - We trained a version of FLAM without sigmoid loss (please see the exact object in link).
>   This version fails to converge and results in substantially degraded SED performance, highlighting the importance of the sigmoid loss in stabilizing training and enabling convergence in the frame-level contrastive setup.
>
> - We trained FLAM using two smaller batch sizes: 256 and 128 (original batch size = 512). As shown in our results, FLAM remains robust across batch sizes, with only marginal drops in performance at smaller sizes. We attribute this to fewer negative examples being sampled during training.
>
> These results support our claim that sigmoid loss is both effective and computationally efficient for frame-level SED. While it is not entirely insensitive to batch size, its formulation allows us to scale to larger batches under the same memory constraints—unlike InfoNCE-based losses, which require global softmax computation and scale poorly with batch size. This makes sigmoid loss particularly well-suited for the resource-intensive frame-level SED setting.
>
> ---
>
> **Typos:**
> Thank you and Reviewer **bqpP** for catching the remaining typos! We've corrected the following:
>
> - **Sec. 3.3**
>   Original: _"we allocate up t slots perup to five times the audio in a batchd pad any unused sloaudio or text ts with placeholders"_
>   **Fixed** (Lines 273–274, Page 5, left column):
>   _“Since each audio clip may contain a varying number of events, we allocate text slots equal to five times the number of audio clips in a batch, padding any unused audio or text entries with placeholders.”_
>
> - **Sec. 5**
>   Original: _"How does FLAM fare on downstream tasks typically used to evaluate contrastive ALMs?"_
>   **Fixed**: _"How does FLAM perform on downstream tasks typically used to evaluate contrastive ALMs?"_
>
> - **Sec. 5.2**
>   Original: _"ustic"_
>   **Fixed** (Line 360, Page 7, left column): _"Acoustic events"_
>
> - **Table 2 (A2T experiment on Clotho):**
>   We fixed the bolding of the correct model.
>
> - **Line 244 (Page 5, left column):**
>   We removed the duplicated phrase: _“the loss $\mathcal{L}_p = $”_

---

> > ### Comment · Reviewer_GGdQ · 2025-04-09
> >
> > Thank you for addressing all my concerns. I'd like to update my score to 4

---

### Official Review · Reviewer_d9oS · 2025-03-13

**Overall Recommendation:** 1

**Summary:**

The paper introduces FLAM, a Frame-Wise Language-Audio Model for open-vocabulary sound event detection. FLAM enhances traditional audio-language models by incorporating frame-level contrastive learning and logit adjustment to handle label imbalance. It leverages a large-scale dataset synthesized from text-labeled audio events, enabling precise event localization.

**Claims And Evidence:**

The claims in the submission are not fully supported by clear and convincing evidence. Key issues include: 1) Lack of supplementary material, which limits reproducibility and detailed analysis. 2) Weak baseline comparisons, as MGA-CLAP is the only baseline re-trained on the same dataset, making it difficult to assess FLAM's true improvements. 3) Limited discussion on the effectiveness of the proposed logit adjustment and data augmentation techniques. These shortcomings reduce the persuasiveness of the claims.

**Essential References Not Discussed:**

No.

**Experimental Designs Or Analyses:**

The paper lacks supplementary material, making it difficult to verify the experimental designs and analyses. The baseline comparisons seem weak, as the primary baseline (MGA-CLAP) is re-trained on the authors' dataset, potentially limiting its effectiveness. More robust baselines and detailed experimental setups would enhance the paper's validity.

**Methods And Evaluation Criteria:**

The proposed methods, including frame-level contrastive learning with logit adjustment and scalable data augmentation, are well-suited for open-vocabulary sound event detection. The evaluation criteria, using both synthetic and traditional SED datasets, effectively assess FLAM's ability to localize events and generalize. However, the baseline comparison could be stronger to better highlight FLAM's advancements.

**Other Comments Or Suggestions:**

1. The paper lacks supplementary material, which could provide additional details on experiments, datasets, and implementation.
2. The baseline comparisons seem weak; including more state-of-the-art models would strengthen the paper's claims.
3. The paper could benefit from a clearer discussion of the limitations and potential future work.

**Other Strengths And Weaknesses:**

Strengths: FLAM introduces a novel frame-wise contrastive objective and logit adjustment for open-vocabulary SED, achieving state-of-the-art performance. The scalable data augmentation pipeline synthesizes a large-scale dataset, enhancing generalization.

Weaknesses: The paper lacks supplementary material, limiting reproducibility. The baseline comparisons are weak, potentially overstating FLAM's advantages.

**Questions For Authors:**

Why did you choose the specific baselines for comparison, and how do you justify the selection given the availability of other state-of-the-art models in the field?

**Relation To Broader Scientific Literature:**

No connection to the broader scientific literature.

**Theoretical Claims:**

The paper does not present any formal theoretical proof. It focuses on empirical evaluations and methodological contributions, such as the frame-level contrastive objective and logit adjustment techniques. Therefore, there are no proofs to verify.

---

> ### Author Rebuttal · Authors · 2025-03-31
>
> We thank reviewer d9oS for reviewing our manuscript and recognizing the novelty of our frame-wise contrastive learning, data augmentation, and FLAM’s strong open-vocabulary SED performance. We respectfully clarify and address the concerns you raised. Notably, these concerns were not shared by the other reviewers. For example:
>
> - **Reviewer Rbvm** commented that “the experiments are fair, with no clear advantages given to the proposed method in comparison to the baseline,” and further stated that “the supplementary material... [is] complete and satisfactory.”
>
> - **Reviewer GGdQ** noted that “FLAM mostly outperforms baselines in some audio-text tasks” and that “the datasets and metrics used in the experiments are standard in this field.”
>
> - **Reviewer bqpP** emphasized that “FLAM builds on previous audio-language models by introducing a frame-level representation, which (as the paper proves) is really important for event detection,” and remarked that “the theoretical analysis and the results... both support [the main claims].”
>
> We hope this provides context for why we believe the raised concerns do not substantially weaken the validity of our contributions. We now respond point-by-point:
>
> >  The paper lacks supplementary material, which could provide additional details on experiments, datasets, and implementation. The paper does not present any formal theoretical proof.
>
> We’d like to clarify that our submission does include detailed supplementary material from page 11 to page 20, covering:
> - A full derivation of the FLAM frame-wise contrastive objective
> - Sound event detection results on open-vocabulary tasks
> - Detailed descriptions of our training procedure, dataset construction, and architectural hyperparameters
>
> In addition, we provide a demo website at [https://flam-model.github.io/](https://flam-model.github.io/), which showcases FLAM's localization results on unseen real-world samples.
>
> Both **Reviewer Rbvm** and **Reviewer GGdQ** explicitly confirmed they reviewed the supplementary materials and found them complete and satisfactory. We invite Reviewer d9oS to revisit this part of the submission.
>
> ---
>
> ### Regarding baseline comparison:
>
> > The baseline comparisons seem weak, as the primary baseline (MGA-CLAP) is re-trained on the authors' dataset, potentially limiting its effectiveness.
> > Why did you choose the specific baselines for comparison, and how do you justify the selection given the availability of other state-of-the-art models in the field?
>
> We chose MGA-CLAP as the primary comparison for open-vocabulary SED because it is the most recent and relevant baseline in this domain. Specifically:
> - **MGA-CLAP** was accepted as an **oral paper at ACM Multimedia 2024** (top 3.97% of submissions), indicating strong peer recognition and technical merit.
> - Its training objective and model design also aim at audio-text alignment, making it a natural benchmark for FLAM.
> - Since our training set comprises licensed proprietary sound effects, we **retrained MGA-CLAP on the same dataset** to ensure a fair comparison. This avoids domain shift effects that would unfairly disadvantage the baseline.
>
> To help readers interpret both in-domain and cross-domain generalization, we also include the original MGA-CLAP performance on public datasets. Additionally, on retrieval and classification tasks, we compare to other baselines like **CompA** and **LAION-CLAP**, where FLAM remains competitive.
>
> As **Reviewer Rbvm** stated: “The experiments are fair, with no clear advantages given to the proposed method.” **Reviewer GGdQ** similarly noted that “FLAM mostly outperforms baselines in some audio-text tasks,” reinforcing that our baseline selection and evaluation strategy are reasonable.
>
> ---
>
> > The paper could benefit from a clearer discussion of the limitations and potential future work.
>
> Thank you for this helpful suggestion. We added a paragraph at the end of Section 7 discussing limitations and future work:
>
> “FLAM represents an initial step toward large-scale open-vocabulary sound event detection, but several aspects remain to be improved. The current training corpus, while diverse, is still limited in scale; future work could explore larger and more diverse corpora, potentially by synthesizing additional labeled mixtures or leveraging web-scale audio. The lightweight model could benefit from scaling or more expressive architectures. Additionally, FLAM uses a fixed 10-second audio input and a coarse frame resolution, which constrains its ability to handle longer or more temporally nuanced recordings. Future efforts could focus on supporting variable-length audio and adopting encoders with finer temporal granularity. Beyond architectural and data improvements, future work could explore the use of real-world frame-level annotations, better evaluation protocols, KL penalty to align frame-level outputs with global model, and generative augmentation strategies to further enhance open-vocabulary localization.”

---

### Official Review · Reviewer_bqpP · 2025-03-18

**Overall Recommendation:** 3

**Summary:**

This paper introduces FLAM, an audio language model, which incorporates a frame-level sound-event localisation loss along with a contrastive learning objective to produce frame-level representations aligned with natural language.

By using a custom augmentation pipeline to combine multiple sounds in a single sample, this paper obtains a training dataset composed of audio, multiple captions, and a binary mask indicating the presence of the captions in the sound at a given frame.

FLAM also compensates for the label imbalance on the dataset (note not all classes have the same duration!) by estimating a per-caption bias and logit-scale.

Experiments on text-to-audio and audio-to-text retrieval show FLAM being competitive with baseline audio-language models (eg. ~86% R@5 for both text-to-audio and audio-to-text retrieval on Clotho compared to LAION-CLAP), while showcasing very strong sound event detection on both open-vocabulary and closed-vocabulary sound event detection tasks.

**Post-rebuttal update**
The main concerns identified during the review process regarded the training set, authors clarified the questions and committed to releasing a validation dataset. Additionally, the authors added experiment ablating the number of frames and the global loss, providing a better understanding of FLAM.

**Claims And Evidence:**

The paper makes the following claims:

**C1. FLAM produces frame-level representations for open-vocabulary event detection**

The theoretical analysis and the results (particularly Table 1) both support this claim.

**C2. FLAM effectively handles label imbalance in sound-event detection training**

This is supported by the overall results (which would not be as good as they are if FLAM could not  handle dataset imbalances) and specifically by Figure 3.

**C3. A pipeline for captioned frame-level audio generation**

The augmentation pipeline is well described in the paper, but key factors are omitted:

 * How large is the dataset?
 * Is every one of the 1M samples 10 seconds long?
 * How many types of events are covered in the captions?
 * How were the proprietary source audio samples licensed? Were they licensed at all??
 * How effective are the dataset augmentation techniques?

**C4. State-of-the-art performance in sound-event detection, strong performance on retrieval and zero-shot classification**

This claim is supported by the results presented on the text.

**Post-rebuttal update**

The authors answered all the questions regarding the training dataset, and committed to releasing the validation dataset ASFX-SED.

**Essential References Not Discussed:**

None that I could find.

**Experimental Designs Or Analyses:**

The experimental analyses that are present seem reasonable.

**Methods And Evaluation Criteria:**

The methods and evaluations make sense for the problem at hand.

**Other Comments Or Suggestions:**

* Line 244 in page 5 (left column) repeats "the loss $\mathcal{L}_p =$"
* Lines 273-274 in page 5 (left column) are corrupted.
* Line 360, page 7, left column, "ustic events" should be "Acoustic events"

**Other Strengths And Weaknesses:**

**Other weaknesses**:

* Missing impact statement.

**Questions For Authors:**

Beyond the questions under **C3** (which I urge authors to address, since they are weighting heavily on my review), I have the following questions:

* **Q1**: Will the training dataset for FLAM be made available? And if so under what terms?
* **Q2**: How large is FLAM? How long does it take it to train? The text states in line 400 (page 8) that LAION-CLAP is larger scale, but it remain unclear by how much.
* **Q3**: What is the effect of varying $L$, the number of frames? Does increasing/decreasing $L$ affect SED/Retrieval/Zero-shot classification performance?
* **Q4**: In table 1, is the gap between `MGA-CLAP*` and `MGA-CLAP (reported)` explained by the different training datasets?
* **Q5**: There seems to be a trade-off between frame representation and global representation (see differences in retrieval scores between `FLAM - Global` and `FLAM` in Table 2). Could this difference be due to model drift when training FLAM with the final loss? And if so, could a KL-penalty term with respect to the predictions of `FLAM - Global` be helpful (similar to the KL penalty used in DPO [1])?


[1] Rafailov et al. (2023) Direct Preference Optimization: Your Language Model is Secretly a Reward Model. NeurIPS.


**Post-rebuttal update**

All these questions were addressed in the rebuttal. Specifically the global loss was confirmed to have an important effect on retrieval (and not much on sound event classification). FLAM with L=128 which performed worse on retrieval/zero-shot accuracy, but similar to FLAM L=32 for sound event classification.

**Relation To Broader Scientific Literature:**

FLAM builds on previous Audio-Language Models by introducing a _frame-level representation_, which (as the paper proves) is really important for event detection.

**Theoretical Claims:**

I did not review the theoretical claims in the papers as those were in the appendix (which I did not have time to review).

I did go through all the derivations in the main text and they seem reasonable. However, as a nitpick, the paper does spend a lot of time justifying the use of a learned per-caption bias term ($\beta^t$), and then figure 3 shows what makes the highest difference is a learned per-caption weight scale ($\alpha^t$) whose use is not justified nor explained in the text (beyond "we experimentally found that it is beneficial").

---

> ### Author Rebuttal · Authors · 2025-03-31
>
> We sincerely thank the reviewer **bqpP** for their detailed and thoughtful review. We appreciate your recognition of FLAM’s contributions and have revised the manuscript to address your comments and questions. Below, we respond point-by-point, reordered for clarity. Due to space constraints, we include additional results and responses on the supplementary webpage:
> **https://flam-model.github.io/response_reviewer_bqpp.html**
>
> ---
>
> ### Questions Related to Dataset
>
> **Dataset size and audio length**
> As noted in Sec. 4.1, the training set includes 1.1M samples. The 1M augmented samples are 10 seconds each, matching FLAM’s fixed-length input (line 317). Original text-audio clips are variable-length; we sample 10-second segments during training.
>
> **How many types of events are covered in the captions?**
> The captions span most categories in the [Universal Category System (UCS)](https://universalcategorysystem.com/). These include nature sounds (e.g., thunder, rainfall, bird calls), urban and human-made sounds (e.g., car engines, speech), and professionally designed sound effects (e.g., gunshots, lightsabers).
>
> **License of dataset**
> All proprietary datasets are fully licensed. We clarified this in the revised opening of Sec. 4.1:
> > “We gather a large mix of licensed proprietary sound effect datasets and ...”
>
> **Will the training dataset for FLAM be made available?**
> Due to licensing, the full dataset cannot be released. However, we will release the ASFX-SED dataset generated by our augmentation pipeline. We hope this will provide a valuable benchmark for future research in open-vocabulary SED.
>
> **How effective are the dataset augmentation techniques?**
> Prior to our work, no training dataset existed for open-vocabulary SED. We show how to scale the construction of such a dataset via data augmentation techniques. This enables frame-level training, since we know frame-level event labels by construction. This makes a crucial difference for significantly outperforming MGA-CLAP (Table 1), which is trained without explicit frame-level supervision.
>
> ---
>
> ### Additional Technical Questions
>
> **How large is FLAM? How long does it take to train?**
> FLAM has ~150M parameters. FLAM-Global trains in ~12 hours; full FLAM in ~24 hours. It shares LAION-CLAP backbones but adds:
> 1. A 1024-dim projection head
> 2. Two MLPs predicting the per-text bias ($\beta^t$) and per-text scale ($\alpha^t$)
>
> In comparison, CompA is larger, using HTSAT-large and FLAN-T5. We updated line 400 to clarify:
> > “Relative to larger-scale ALMs like CompA, FLAM remains competitive, particularly on VGGSound.”
>
> **Effect of varying L (number of frames)**
> We thank the reviewer for this suggestion. Due to limited time, we trained a variant of FLAM with L=128 and observed a trade-off between SED performance and retrieval/zero-shot performance. See the supplementary link above for results.
>
> **In table 1, is the gap between MGA-CLAP\* and MGA-CLAP (reported) explained by the different training datasets?**
> Yes, this gap is due to dataset differences. MGA-CLAP was trained on large-scale web data, including YouTube, which aligns with DESED and AudioCaps. MGA-CLAP* was retrained on our proprietary dataset, explaining the lower performance.
>
> **There seems to be a trade-off between frame representation and global representation [...]. Could this difference be due to model drift when training FLAM with the final loss? And if so, could a KL-penalty term with respect to the predictions of FLAM - Global be helpful (similar to the KL penalty used in DPO [1])?**
> Excellent observation! Indeed, your observation reflects a trade-off between local and global representation alignment. To investigate further, we conducted additional experiments training FLAM without the global loss. Removing global loss results in a slightly better SED performance but a significant drop in retrieval and zero-shot performance. Please refer to the results in the link above. We mitigate this trade-off via joint global contrastive optimization. As the reviewer suggested, incorporating a KL penalty to align frame-level outputs with the global model is a promising direction. We’ve noted this in the discussion for future work.
>
> **The impact and intuition about logit scale**
> Intuitively, a smaller logit scale increases the cosine distance between negative frame and text embeddings for the same loss effect. This helps the model capture finer distinctions in cosine similarity. Due to space constraints, please refer to the link above for more discussion.
>
> ---
>
> ### Other Comments
>
> **Missing impact statement**
> Many thanks for pointing this out. We have added an impact statement at the end of the paper. See the link above for the updated version.
>
> **Typos**
> We thank the reviewer for highlighting the typos—these have been corrected. Due to space constraints, full details are in our response to reviewer **GGdQ**.

---

> > ### Comment · Reviewer_bqpP · 2025-04-07
> >
> > Thank you for the additional experiments and detailed rebuttal. Based on the clarifications for C3 as well as the ablations without global loss and L=128, I have decided to raise my score to 3.

---

> > > ### Author Response · Authors · 2025-04-07
> > >
> > > Dear Reviewer bqpP,
> > >
> > > Thank you very much for your detailed and constructive review of our paper. We sincerely appreciate your thoughtful feedback, insightful questions, and the time you took to engage with both the main submission and our rebuttal. Your comments on the dataset, model design, and the local-global trade-off were especially helpful in improving the paper. We're grateful for your updated score and support.

---

### Decision · Program_Chairs · 2025-05-01

**Decision:**

Accept (poster)

**Comment:**

This paper introduces an audio language model ("FLAM") utilizing a frame-level sound-event localisation loss along with a contrastive learning objective to produce frame-level representations aligned with natural language. Authors develop open-vocabulary data augmentation (with localization) techniques with binary masks to indicate the presence of the captions in the sound at a given frame. FLAM also compensates for the label imbalance on the dataset (not all classes have the same duration!) by estimating a per-caption bias and logit-scale.

Experiments on text-to-audio and audio-to-text retrieval show FLAM being competitive with baseline audio-language models (eg. ~86% R@5 for both text-to-audio and audio-to-text retrieval on Clotho compared to LAION-CLAP), while showcasing very strong sound event detection on both open-vocabulary and closed-vocabulary sound event detection tasks. During the rebuttal, authors clarified several questions regarding the training process and data and committed to releasing the ASFX-SED validation dataset, which should be a useful resource to the community. Ablations confirmed that the global loss has an important effect on retrieval (and not much on sound event classification), leaving reviewers that engaged in the rebuttal process with a good sense of how the model works and an accept recommendation.